# A Novel on Transmission Line Tower Big Data Analysis Model Using Altered K-means and ADQL

**Se-Hoon Jung** [1] and **Jun-Ho Huh** [2,*]

[1] School of Connection Major (Bigdata Convergence), Youngsan University of Yangsan, Yangsan 50501, Korea
[2] Department of Software, Catholic University of Pusan, Busan 46252, Korea
* Correspondence: 72networks@pukyong.ac.kr or 72networks@cup.ac.kr

**Abstract:** This study sought to propose a big data analysis and prediction model for transmission line tower outliers to assess when something is wrong with transmission line tower big data based on deep reinforcement learning. The model enables choosing automatic cluster K values based on non-labeled sensor big data. It also allows measuring the distance of action between data inside a cluster with the Q-value representing network output in the altered transmission line tower big data clustering algorithm containing transmission line tower outliers and old Deep Q Network. Specifically, this study performed principal component analysis to categorize transmission line tower data and proposed an automatic initial central point approach through standard normal distribution. It also proposed the A-Deep Q-Learning algorithm altered from the deep Q-Learning algorithm to explore policies based on the experiences of clustered data learning. It can be used to perform transmission line tower outlier data learning based on the distance of data within a cluster. The performance evaluation results show that the proposed model recorded an approximately 2.29%~4.19% higher prediction rate and around 0.8% ~ 4.3% higher accuracy rate compared to the old transmission line tower big data analysis model.

**Keywords:** altered K-means; A-Deep Q Learning; big data analysis; transmission line tower big data; artificial intelligence; reinforcement learning; machine learning; Python

---

## 1. Introduction

In recent years, the application of big data analytics models in various industrial/service/ technical sectors, including financial/banking [1,2], health care [3,4], Internet of Things (IoT) [5,6], communication [7,8], smart cities [9,10], and transportation [11] has resulted in many significant innovations supporting market growth in their respective systems. As one of the notable changes in the period of transition from the existing power grids to a smart grid, much more data are continuously generated from localized systems/data integration, controls, or applications. These form big data that can be used to design a more efficient smart grid for the future [12]. The modern power grids adopt a series of innovative elements of ICT, system control, and measurement technologies to operate and maintain their respective power systems for the provision of reliable and sustainable energy. Such power grids often have broad-area Advanced Metering Infrastructure (AMI) constructed based on the network of smart meters and Phasor Measurement Units (PMUs) that allow the power system itself to take high-precision electrical measurements [13–15].

When big data created through AMI are utilized together with some external data, such as usage patterns, weather conditions and demand supply statuses, etc., these power grids will become more intelligent. At the same time, they provide quality power by allowing the power system to check consistently the grid's condition and user patterns as well as for any possible system flaws and providing alternative solutions in some cases. Such huge heterogeneous data combination [16–20] can

be quite effective for the power grid when used along with the data obtained from the operational simulations using them. In general, a huge volume (terabytes) of data flows into these modern or smart grids annually [21–29] through various types of power network elements (e.g., sensors, smart equipment, etc.) that constantly collect the data generated by geographic information systems, SCADA systems, or weather and traffic information systems or from social networks [17–20]. The development of information and communication technologies based on the advance of hardware and software creates new values across various fields of society in conjunction with the Big Data era [30]. In today's world, where the data generation cycle is becoming increasingly short, it is necessary to conduct ongoing research on the classification and analysis of internal data as well as hidden patterns in various phenomena data. In energy-related industries, in particular, research on smart electric power data to combine information and communication technologies with transmission line tower data has become indispensable [31–33]. As the consumption of electric energy has increased across various industrial fields around the world, there is an absolute need to supply transmission line tower and analyze transmission line tower systems in a stable fashion. A stable transmission line tower analysis system definitely requires analysis from a supply perspective and analysis and management from the demand perspective [33].

Big data are expected to provide a series of great opportunities for the existing power grid system, yielding profitable returns in society and economy. Nonetheless, the question of how to apply big data to the power grid operation planning effectively should still be studied further, requiring in-depth research on its application methods and effectiveness. The big data that can be applied to a power grid may consist of some disparate data, as they are collected from various fields using different measurement or communication technology. Applying such data to power grid operation planning or to the power system itself to achieve efficient, secure grid operation requires meticulous study on data security and issues associated with the complexities involved in the process of data computation, classification, and integration. As such, maintaining the modern power grids, which are becoming increasingly complex and are required to provide flawless service, requires the development of a more advanced methodology for applying big data. Thus, a big data analytics model with a grid system operation visualization function will be quite useful when attempting to improve comprehensive operational awareness or making present and future decisions.

Many different data analysis methods have been developed, including supervised learning, unsupervised learning, and reinforcement learning algorithms. Electric power data analysis models usually analyze continuous and categorical electric power data first including linkage, correlation, regression, and time series analysis among supervised learning techniques. They generate learning data based on the analyzed data and learn new data inputs. Time series analysis makes use of time-related features such as self-regression models, moving average models, ARIMA models, trend factor disintegration, season factor disintegration, circulation factor disintegration, and irregular factor disintegration. Correlation coefficients are used to analyze relations and influences between two kinds of data based on the analysis of internal factors of the collected transmission line tower big data. As a technique used to analyze the internal characteristics of the collected transmission line tower big data, clustering involves forming a cluster of similar data of similar nature through the connectivity of data. Recently, research has been conducted on a transmission line tower analysis system through the clustering technique as part of cluster analysis [34–42]. The clustering technique is especially characterized by classification and analysis based on the detection of outliers according to sensitivity among clusters and discovery of problems with transmission line tower data according to outliers. There has been an ongoing need for research to analyze the data of outdoor transmission line towers in the domain of internal power IoT data analysis. Transmission line towers form a power supply line along the streets, recently causing several issues worldwide including spontaneous combustion and fire. In Republic of Korea, a spontaneous combustion case happened in Gangneung-si in April 2019 and caused huge damage to human life and property. Had there been an analysis model for transmission line tower data, it would have helped reduce the extent of damage. Since transmission line towers

are all equipped with sensors, it is easy to collect and analyze their data. Note, however, that they were classified as power data managed by the government, which makes it difficult to secure the data. Transmission line tower data consists of the following: transmission line towers, pillar-shaped devices, wires, insulators, underground structures, ground transformers, ground switches, underground cables, atmospheric environments, temperature, humidity, acceleration, sound detection, ultrasonic waves, light detection, terrestrial magnetism, current, pressure, and contamination. Correlations between internal data are analyzed by investigating the relations between large amounts of energy data and checking the influences between them. Outliers and novel sections are identified by setting a hypothesis to be tested and the relation between the tools and materials of transmission line towers and data, for use as learning data to take countermeasures for future transmission line tower IoT data issues.

There are various problems with supervised and unsupervised learning used in previous studies. First, as a technique of supervised learning to increase analysis efficiency, correlation analysis should include a method of estimating and testing the relational expression after predicting relations based on dispersion in order to analyze non-linear internal data. Another problem with using correlation analysis research in analyzing transmission line tower IoT data is the difficulty of a certain data item to influence two data items in a complex way. Third, unsupervised learning should regulate sensitivity based on hypothesis testing and clustering and facilitate categorizing outlier data in order to analyze transmission line tower IoT data. Finally, the old models for checking for problems with electric data have a data structure of checking outliers based on standardized or labeled data. When different data items are added from the old ones, they can lower the accuracy and prediction rate of data analysis. This study proposes an analysis and prediction model for transmission line tower outlier data to assess problems with transmission line tower big data based on an altered K-means algorithm that supplements the K-means algorithm according to non-labeled sensor data and deep reinforcement learning for self-study. The old K-means algorithm divides outliers into data distributed in and outside the cluster boundary by designating data sensitivity in advance and checks data randomly. As proposed in this study, the transmission line tower big data clustering model analyzes problems with the old K-means algorithm, chooses automatic cluster K values, and contains transmission line tower outlier data. The transmission line tower big data clustering architecture basically contains a new algorithm to compensate for the problems with the K-means algorithm. A data clustering architecture is designed in three major levels: the row data level to collect and pre-treat transmission line tower big data; the clustering level to apply an altered K-means algorithm; and the reinforcement learning level to check clustered transmission line tower data outliers and promote self-study. The row data level consists of the processes of collecting data, eliminating unnecessary data, and normalizing data. The clustering level includes the altered K-means algorithm proposed in this study. The reinforcement learning level features an outlier learning model of low-level clustering to apply A-Deep Q-Learning, which changes the off-policy and Q-table methods.

The rest of this paper is organized as follows: Chapter 2 introduces previous studies on the old K-means algorithm for data classification, reinforcement learning, and transmission line tower big data analysis models; Chapter 3 proposes an altered K-means algorithm to assess outliers with transmission line tower big data and describes a reinforcement learning model of A-Deep Q-Learning to increase the efficiency of transmission line tower big data analysis; Chapter 4 describes the policy simulations performed to examine the usability of the proposed A-Deep Q-Learning; Chapter 5 discusses the experiments and performance evaluations carried out for the proposed technique; Finally, Chapter 6 presents the content and findings of this study and proposes future research plans.

## 2. Related Research

### 2.1. K-Means Algorithm

The K-means algorithm is a clustering technique used to classify input data into k clusters based on unsupervised learning similar to supervised learning [43–49]. Unlike supervised learning, which

updates weight vectors every time a vector is entered, the K-means algorithm updates weight vectors simultaneously after all input vectors are entered. Algorithm 1 shows base K-Means Algorithm. The criteria of clustering classification are the distance between clusters, dissimilarity among clusters, and minimization of the same cost functions. The similarity between data objects increases within the same clusters, but that to data objects in other clusters decreases. The algorithm performs clustering by setting the centroid of each cluster and the sum of squares between data objects and distance as cost functions and minimizing the cost function values to repeat the cluster classification of each data object.

---

**Algorithm 1** Base K-Means Algorithm.

① The user will determine K, the number of clusters, in advance.
② One of the entire group of entities will be included in each of the clusters in the determined number of K.
③ All the entities for clusters will be assigned to the center of the closest cluster according to the distance-based method.
④ After the process in ③, the center of the assigned entities will be set as the new central point of the concerned cluster concerned, which will then be re-assigned.
⑤ Stages③ and ④ will be repeated until there is no more migration of entities.

---

The K-means algorithm is a clustering technique used to classify input data into k clusters based on unsupervised learning similar to supervised learning [43]. Unlike supervised learning, which updates weight vectors every time a vector is entered, the K-means algorithm updates weight vectors simultaneously after all the input vectors are entered. The criteria of clustering classification are the distance between clusters, dissimilarity among clusters, and minimization of the same cost functions. Similarity between data objects increases within the same clusters. Similarity to data objects in other clusters decreases. The algorithm performs clustering by setting the centroid of each cluster and the sum of squares between data objects and distance as cost functions and minimizing the cost function values to repeat cluster classification of each data object. Intra-cluster distance (IntraCD) is the addition of distance to all the input vectors allocated to the concerned cluster from the centroid of each cluster. Inter-cluster distance (ICD) is the distance of weight vector between two clusters. As seen in Equation (1), errors are calculated by adding the sum of IntraCD of all the clusters and subtracting the sum of ICD for all the cluster pairs. β and γ are weighted values.

$$\text{Error} = \beta \sum_{i=0}^{k}(Intra\ CD) - \gamma \sum_{i=0}^{k}(ICD), \tag{1}$$

MacQueen Approach K-means [MAK] presents a method that is usually used in research on the utilization of the K-means clustering algorithm [50]. The user arbitrarily selects K, the initial number of clusters, from the initial objects for clustering. K can be randomly selected from the entire group of objects and is assigned to the closest cluster according to the measurement of distance from the cluster centroid to all of the objects. The objects assigned to a cluster are finally used for reassignment to new centroids. The algorithm is terminated when centroids are measured to be below the threshold value set by the user. Kaufman Approach K-means [KAK] introduces an algorithm to curb the rising costs of measurement due to the calculation of distance from all of the objects, which had been pointed out as a problem with MAK [51]. Faster in measurement than MAK, this algorithm determines the centroids of all the objects distributed as initial centroids. The distance is measured from an object of the entire group to the initial cluster value. When a selected object has a distance from the initial value over the threshold value set by the user, with a certain number of objects nearby or more, it is determined as a cluster centroid. The algorithm repeats the process until the initial K value matches the selected outcome up to termination. Note, however, that the approaches of MAK and KAK to initialization have problems, i.e., they select initial values arbitrarily and choose initial values according to certain conditions and rules. Max-min Approach K-means [MMK] selects an object from the entire group, determines it as the first initial value, and measures its distance from the other objects [51]. The object with the farthest distance from the first initial value is assigned as the second initial value. The distance from the other objects is calculated based on the first and second initial values. Since two initial values

are needed to calculate distance, the sets of distance pairs are measured to save two measurements. The initial value of distance pairs from the measured set of distance pairs is defined as the distance measured between the object concerned and two values. The observed value with the maximum distance measurement is determined as the third initial value, which has the longest distance from the earlier two initial values assigned. The algorithm continues until the initial values of K are all met through the repeated process. K-means++ [KMP] selects an object from the entire group arbitrarily and assigns it as the first initial value [52]. Distance is measured from the first initial value to the entire group of objects. Each measurement is converted into squared distance and divided by the squared addition of distance of all the objects to calculate the probability of selecting the second initial value. Starting with the third initial value, the distance from the other objects is calculated using the same method as MMK. The probability calculation is then repeated to assign initial values [53].

## 2.2. Reinforcement Learning

Although there are differences in the academic background of reinforcement learning across various fields, most academic fields apply a decision-making algorithm to obtain optimal results. For instance, the reward systems of neuroscience, game theory of economics, optimal control of engineering, and applied science of math and industrial engineering maximize the rewards for the selected action through learning. This is called reinforcement. Reinforcement learning handles this issue with the concept of reinforcement. "Reinforcement" can be understood based on the Skinner box experiment in behavioral psychology. In a box with reward and punishment set for certain actions, for instance, an animal in the experiment learns the conditions of reward and punishment by making an attempt at many actions and gradually choosing the action that will result in reward. In reinforcement learning, one has various experiences, checks the outcomes of the experiences, and tries to act in ways that produce better outcomes. This attempt is repeated to acquire a method of fulfilling a goal. Reinforcement learning is a method of learning how to act in a given situation to maximize reward for the action. The old machine learning model expresses state changes for action in the first law's physical formula or statistical/probabilistic relations, whereas reinforcement learning does not define the relations between action and state in the learning process. The learning methods of reinforcement learning, called artificial intelligence, are similar to those of humans, attracting attention from many sectors in recent years [54–70].

As the most used algorithm in reinforcement learning [54], Q-Learning is a reinforcement learning algorithm that is capable of learning without an environment model. It has achieved successful results in many studies, and its purpose is to learn Q-Function for an agent to get maximum results. In deep reinforcement learning, an agent learns with Deep Q-Network combining deep artificial neural network technologies with Q-Learning without depending on domain knowledge. Since a system is dependent on domain knowledge, it can operate well only in certain domains, and it has difficulty expanding to other domains, hence the need for technologies that are not dependent on domain knowledge. Deep reinforcement learning (DRL) can solve these problems [61]. The A3C algorithm does not requires a large memory capacity, which is a huge disadvantage of the old DQN, which uses replay memory to save gameplay data for the stabilization of learning [62]. A3C solves this problem by implementing multiple independent environments in parallel instead of replay memory and collecting learning data in each environment. It is critical to strike a proper balance between inquiry and utilization in the field of artificial intelligence algorithms, as well as reinforcement learning. In deep reinforcement learning, inquiry and utilization remain difficult problems to solve. The deep reinforcement learning algorithms investigated recently are less dependent on domain knowledge, solving various problems at the human level and exploring wide inquiry space, thus adding more efficient inquiry techniques than previous studies to their research list [63–68]. Simple continuous actions are generally referred to as the bundle of actions needed to achieve a sub-goal rather than options. Sub-goals refer to the middle stages needed to achieve the final goal. If an agent gets a reward at every middle stage, he or she will be able to continue inquiry toward the final goal more easily. Unlike the final goal, whose

definition is clear, one should depend on domain knowledge to distinguish and use sub-goals. When sub-goals are defined incorrectly or with insufficient analysis, they can cause deceptive problems. Some studies have identified a confidence interval based on the number of attempts at action in certain conditions and have used it in inquiry [63,64]. Others have identified characteristics from state changes to ensure the continuous movement of multiple agents and have used them in inquiry [65]. This is called pixel control or feature control. Still others have conducted research on the improvement of inquiry performance by using additional information about distance and objects in an environment. The most fundamental way to solve the issue of insufficient inquiry performance is to predict the future state with an explicit planning algorithm [66]. When implementing a complex procedure, human beings work according to plans and procedures rather than reactive processing. There are some planning algorithms applying this procedure [67]. Some research studies have been conducted on the utilization of the knowledge acquired in a previous environment in a new and different environment based on PathNet by combining evolutionary computation with deep reinforcement learning [68]. Note, however, that these reinforcement learning algorithms and inquiry and utilization algorithms are dependent on large amounts of learning data and domain information. As a result, they depend on hardware and incur large computation costs [69–76].

## 2.3. Electric Power Prediction System

Various research studies have been conducted on a prediction system for power-related data [77–82]. In recent years, active research efforts have been made to investigate pattern mining such as power demand and patterns and analyze outlier for the identification of defective data from the collected data [83,84]. This study compares and evaluates the model proposed in the three research studies below to analyze the transmission line tower sensor data of power-related data [85–87].

D.I. Park, et al. [85] proposed clustering and classification analysis and performance evaluation to analyze the patterns of electric consumption based on the time series data of time units. The research used cluster analysis to identify the types of electricity demand and utilized classification analysis for determining the optimal numbers of clusters. It considered external factors including temperature, precipitation, wind velocity, humidity, amount of sunshine, and holidays for the classification analysis. It also presented the results of classifying electricity demand patterns using four classification methods—decision-making tree, random forest, SVM, and Naive Bayes—and proved the superior performance of a classification method based on the random forest technique. It had the advantage of predicting mean electricity demand patterns throughout the day based on data from the Meteorological Office in the random forest method. When choosing the optimal number of clusters, however, it used the heuristic method based on the accuracy of classification analysis and selected the number of clusters through repetitive learning instead of automated optimal cluster, thus increasing the analysis costs.

S.H. Ryu, et al. [86] made use of clustering with respect to the daily and annual load patterns for electric consumption data using various clustering methods and analyzed their connections with KSIC labels. Ward linkage-based hierarchical clustering was used to treat cluster analysis for clustering. The electric load patterns were analyzed daily and annually. Since the pattern data showed a linear, continuous distribution, density-based DBSCAN was eliminated from the analysis. Note, however, that the study did not reflect outlier detection by the DBSCAN algorithm even though it was more appropriate for identifying outliers in the distribution of linear continuity than the hierarchical clustering technique.

J.H. Shin, et al. [87] proposed an analysis model to analyze equipment load changing every moment under the line at every 15 minutes by installing no measuring devices in large-scale electric facilities and using meter data obtained from all clients every 15 minutes or every month. It connected the multiple operational systems needed to calculate and test load, obtained remote and monthly readings and transformer and line load, and built a model data warehouse for pattern analysis based on the data. Clustering, classification, and time series data mining techniques were also used to predict load patterns.

## 3. Proposed Transmission Line Tower Analysis Algorithm

### 3.1. Structure of the Proposed System

Figure 1 shows the block diagram of the proposed system. The proposed analysis model for transmission line tower data consists of three levels: the row data level for transmission line tower data preprocessing, including transmission line tower data collection and preprocessing; the clustering level to which an altered K-means algorithm would be applied; and the reinforcement learning level to learn how to check for outliers in clustered transmission line tower sensor data for itself. The row data level involves gathering transmission line tower data, eliminating unnecessary data, and normalizing data. The clustering level includes an altered K-means algorithm to which the principal component analysis proposed in this study would be applied. The reinforcement learning level features a learning model for predicting outlier in low-level clustering to apply A-Deep Q-Learning altered from the off-policy and Q-table methods.

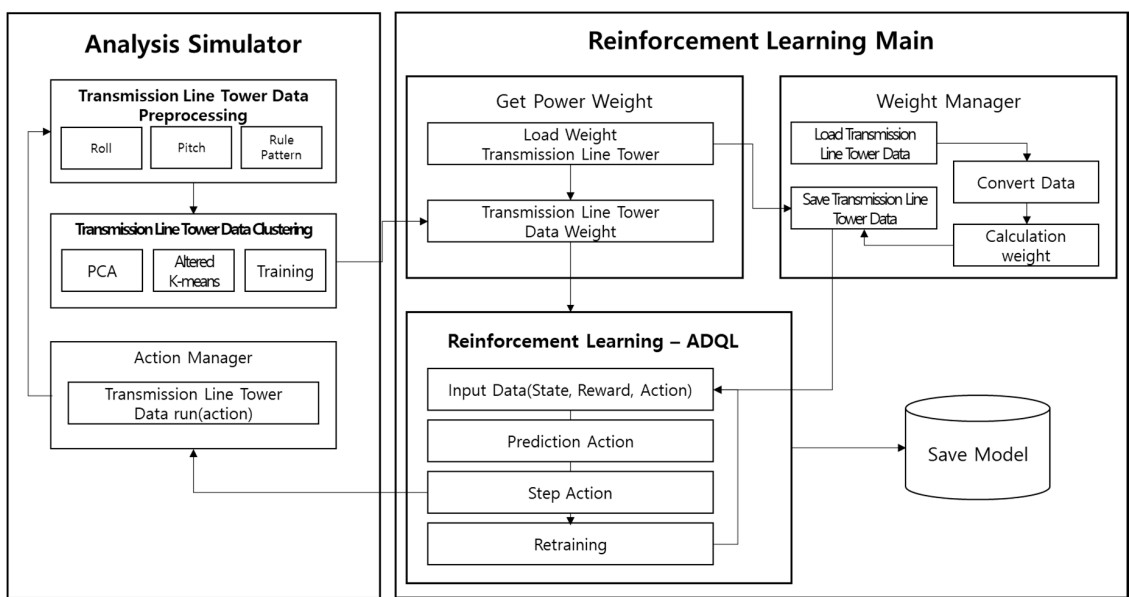

**Figure 1.** Structure of the proposed analysis system.

### 3.2. Raw Data Level

In this study, the raw data level included data collection, elimination of unnecessary data, and normalization of data to analyze transmission line tower data. The collected sensor node data offered information about the number of transmission line towers, location of equipment, temperature, humidity, pitch, roll, intensity of illumination, ultraviolet light, pressure, remaining battery, and cycle. The collected raw data went through the processes of eliminating unnecessary data, identifying only certain columns, and normalizing data. The data normalization process digitized the data scope within the available scope and re-processed it into a numerical scope for application to an analysis model.

### 3.3. Clustering Level : Altered K-Means Algorithm

This study used an unsupervised learning-based altered K-means algorithm [33,53] to analyze pre-treated transmission line tower IoT sensor data. Note, however, that K-means algorithms have the disadvantage of lacking a data classification structure that has been optimized for application in Big Data analysis. In terms of the conditions of Big Data creation, including variety, volume, and velocity, the old K-means algorithms have limitations in data classification with regard to volume and velocity. The transmission line tower IoT sensor data covered by this study were characterized by the fast formation and change of volume and velocity, The transmission line tower IoT sensor data covered by this study were characterized by the fast formation and change of volume and velocity,

which means that there were many problems with the application of the old K-means algorithm in this study. Thus, an altered K-means algorithm-based analysis model was proposed. Since transmission line tower IoT sensor data contain Big Data forms, cluster classification based on the old K-means algorithm can have low accuracy. This study applied an automatic K-value selection algorithm for the clustering of massive transmission line tower IoT sensor data and classified the sensor node data of transmission line towers first. In altered K-means, K values were determined based on the reduced dimension of principal component analysis in the input of multidimensional transmission line tower IoT sensor data. The reduction of dimensions was designated within the scope that could explain 90% of all the data. The scope of principal components, including up to 90%, was selected based on K the number of clusters in K-means. In the principal component analysis, covariance helps check the clusters selected from the given transmission line tower data and their connections with other clusters. Based on this information, the cluster boundary can be distinguished through principal component analysis. The principal component of cluster indicator vector is defined as $v_i$ in Equations (2) and (3). Equation (4) defines the scope of meeting the optimized conditions of the initial central value.

$$k_1 = \{i \big| v_1(i) \le 0\}, \; k_2 = \{i \big| v_1(i) > 0\} \tag{2}$$

$$n\overline{y^2} - \lambda_1 < K_{k=2} < n\overline{y^2} \tag{3}$$

The mean distance between the data within a random cluster and the randomly selected central point were determined based on the addition of Euclidean squared distance. Equation (4) represents the mean distance between a randomly selected central point and all the input data. $S(k)$ of Equation (7) is based on the difference between $A_k$(cohesion) as the mean distance from data inside a random cluster in Equation (5) and $B_k$(separation), which is the mean distance from data outside a random cluster in Equation (6). As the optimal dissimilarity of the cluster of the maximum value, $S(k)$ becomes K the number of clusters as in Equation (8). $S(k)$ is between −1 and 1. The one closer to 1 is chosen as the optimized number of clusters.

$$A_k = \sum_{k=1}^{k} \sum_{i \in C_k} (X_i - m_k)^2 \tag{4}$$

$$B_k = \min\{(X_i - m_k)\}^2 \tag{5}$$

$$S(k) = \frac{1}{N} \frac{B_k - A_k}{\max(A_k, \, B_k)} \tag{6}$$

$$S(k) = \begin{cases} 1 - \dfrac{A_k}{B_k}, & A_k < B_k \\ \dfrac{A_k}{B_k} - 1, & A_k > B_k \end{cases} \tag{7}$$

A sequential space division algorithm was applied to transmission line tower IoT sensor input data instead of the old algorithm-based random selection, in order to increase clustering accuracy and efficiency and select the initial central value of repeating clusters. In the selection of the initial central point, the locations of data were used in the distribution and standard normalized distribution of similarity and density. Of the entire input vector data, $m_k$, the central point of the first cluster $k_1$, is distributed within the scope of $P(\overline{X} \ge x_k y_k) = \pm 0.9$ or higher based on the standard normalization distribution $\varnothing_{\mu,\sigma^2}(x_k y_k)$, hence its classification as an observed value item for the initial central point. The observed value that will be the initial cluster central point along the standard normalization distribution is $k^2$ or lower, and the conditions concerned include $X$ as the set of input data, the mean µ, and standard deviation σ. When K, the number of clusters, is 0, clustering will proceed as in Equation (9).

$$\varnothing_{\mu,\sigma^2}(x_k, y_k) = \left( \frac{1}{\sigma\sqrt{2\pi}} e^{-\frac{(x-\mu)^2}{2\sigma^2}} \right) \times \left( \frac{1}{\sigma\sqrt{2\pi}} e^{-\frac{(y-\mu)^2}{2\sigma^2}} \right) \tag{8}$$

$$P\big(|x - \mu| \geq k\sigma\big) \leq k^2$$
$$P\big(|x - \mu| \geq k\sigma\big) = P\big((x - \mu)^2 \geq k^2\sigma^2\big)$$
$$= \frac{k^2\sigma^2}{E\big[(x - \mu)^2\big]}$$
$$= \frac{k^2\sigma^2}{\sigma^2}$$
$$= k^2 \tag{9}$$

The equation to measure an object distance $A_k$ will be defined as in Equation (10) to measure similarity and density among objects in case of one object or more in the observed values of centroids. The mean distance measurement between objects, $AVG_k$, will be defined as in Equation (11).

$$A_k = d(x_{ki}, x_{li}) = \sum_{k=1}^{k} \sum_{i \in C_1} (X_i - X_k)^2 \tag{10}$$

$$AVG_k = \frac{1}{k} \sum_{k=1}^{k} \sum_{i \in C_1} (X_i - X_k)^2 \tag{11}$$

$A_k$ as the distance between data is measured for $x_i$ for all the data except for the central point based on $m_1$ as the central point of the first cluster $k_1$. The data of the maximum measurement is attached to $m_2$ as the central point of the second cluster $k_2$. It is defined as in Equation (12).

$$C_2(m_2) = \alpha_i \leftarrow \max_{1 \leq i \leq n}(A_k)$$
$$= \max_{1 \leq i \leq n}(A_k) \parallel x_i - m_1 \parallel$$
$$= \parallel \alpha_i - m_1 \parallel \tag{12}$$

Algorithm A1 in Appendix A shows the altered K-means algorithm proposed in this study for the clustering of unlabeled data [30,33,39,44,53].

### 3.4. Reinforcement Learning Level: A-Deep Q-Learning Algorithm

The return value obtained through the state and action value function of reinforcement learning becomes an important criterion for determining an action in a given state; it is used as a policy for reinforcement learning. The determination of value function value is that of reinforcement learning policy. There are various methods of exploring value function values in reinforcement learning. The deep Q-Learning used in this study explores policies based on experiences with no definition of environment, and it can be effectively applied to many states and actions faced in reality. In this study, reinforcement learning was aimed at learning the outliers of clustered transmission line tower data and checking the outliers and singularities in outlier-related data. They were treated in the standard organization of reinforcement learning consisting of outlier data interacting with Environment which was a learning and distribution environment for the clustered prediction data of transmission line tower outliers in the discrete time step. A good example is Q-Learning. In this study, deep learning was applied with a neural network used to obtain action numbers. It was designed to increase learning performance considerably and achieve a human-level success rate. The Markov Decision Process (MDP) was applied to the problem-solving algorithm expressed in MDP based on the transmission line tower data classified according to the clustering results of transmission line tower data. Equation (13) shows the process.

$$\Pr R_{t+1} = r, \ S_{t+1} = S'|S_t, \ A_t| \tag{13}$$

In this expression, S represents the clustering results of transmission line tower, A, the clustering outcomes of transmission line tower data, and R, the reward for the clustering outlier results. When obtained, the outliers of electric data clustering are accompanied by rewards. In this study, rewards were obtained when the number of repeated outlier measurements was two or more. At the end of

each episode, all the rewards from the beginning were added. Equation (14) shows the addition of all the rewards.

$$A_t = R_{t+1} + rR_{t+2} + rR_{t+3} + \ldots = \sum_{k=0}^{\infty} \gamma^k R_{t+k+1} \tag{14}$$

On the other hand, $\gamma$ represents a discount factor, i.e., the influence of future reward on the present. As the steps were completed, the reward value of the past was closer to 0. The clustering containing outliers received the same reward. In Equation (15), Q-Learning is a type of reinforcement learning and algorithm for choosing the optimal state based on the reward information according to action. In each step, action is determined according to the Bellman optimization equation in a value function.

$$Q(s, a) \leftarrow R + \gamma maxQ(s', a') \tag{15}$$

Q-Learning understands the environment of transmission line tower data clustering outliers in the Markov decision process, saves the action-value function for all the states in table form, and proceeds with learning by applying it. In this study, a separate target network learning method for refresh buffer and $x_t$ calculation was applied to Q-Learning. The target value of reward learning is found in Equation (16), whereas Equation (17) shows a loss function. In Equations (16) and (17), $\theta^Q$ represents the weight of outliers, and $\theta^Q$, the weight of outlier accumulated twice or more. The reward value for action was calculated according to the burden of each action on hardware. Algorithm 2 shows the reward according to action.

$$x_i = \gamma_i + \gamma Q'\left(s_{i+1}, \mu'\left(s_{i+1}\middle|\theta^{Q'}\right)\right) \tag{16}$$

$$U = \frac{1}{N}\sum_i \left(y_i - Q\left(s_i, a_i\middle|\theta^Q\right)\right)^2 \tag{17}$$

---

**Algorithm 2** Reward Function Algorithm

**if** Clustering and subClustering distancet−1 > Clustering and subClustering distance *t*
    **then**
        reward=0.5
**else**
    **then**
        reward=−0.5
**if** action == 0
    **then**
        reward = reward + 0.2
**else if** action == 1 **or** action == 2
    **then**
        reward = reward − 0.2
**else if** action == 3 **or** action == 4
        **then**
        reward = reward − 0.3
**if** outlier
    **then**
        reward = −1.2
**else if** goal
    **then**
        reward = 1.2
**return** reward

---

Algorithm 2 shows Reward Function Algorithm. Since transmission line tower outlier data should be included in a cluster for reward, the progress of learning episodes should achieve shorter distance to the cluster center and shorter time to it. Despite the development and excellent performance of DQN, however, there is only one action value for a discrete environment. It means that there is no way of knowing the environment including the data and action around a cluster other than the information

about the cluster center. In an effort to solve this problem, this study saved the distance value of action between data inside a cluster or Q-value as the network output used in the old DQN in external memory. It was then turned into a new state value, adding information to recognize the distance between the cluster center and data. This approach has the advantage of predicting and learning the next action based on the memory of the previous action (distance inside the cluster containing outliers) without separate cluster modeling. This technique helps check the decision about outliers of external data through reinforcement learning without information about the side of a cluster containing outliers. Actions can be taken according to the decision of outliers in the input data based on the action history of discrete actions by t (determination of items in a cluster containing outliers). Equation (14) shows A with n number of A-Deep. In this study, a policy of making a fast approach to transmission line tower outlier data according to the learning goals of DQN was implemented. In a queue structure, A-Deep has the old action disappearing when a new action value is saved. A-Deep is saved in a refresh buffer and used in mini-batch learning. Another problem of DQN is the overestimation of action that can be selected in a given state. To solve this, this study made the main neural network choose actions as in DDQN and the target neural network create a target Q-value for the action. In Equation (15), the number of target networks was updated to reduce the overestimation of certain actions and ensure fast learning.

A total of five fully connected hidden layers in the artificial neural network were used in reinforcement learning. The ReLU activator was used in the network except for the output layer for transmission line tower outlier data of 0~5. The ReLU activator, which became famous through deep learning, mitigates the vanishing gradient phenomenon of a deep neural network. Each of the layers had the same 20 neurons. Inputs consisted of the clustering results of transmission line tower data through altered K-means and distance between clusters. Additionally, batch normalization was applied, becoming almost an essential step to reduce the burden on the computer in recent deep learning studies. The Xavie technique was applied to each weight, determining an initial value only with the number of inputs and outputs. In network approximation, the loss function was reduced using the Adam optimizer, which added a momentum algorithm to the old gradient decent optimizer, instead of the gradient decent optimizer that could easily fall into local minima.

Algorithm A2 in the Appendix A shows the ADQL algorithm as a reinforcement learning algorithm for the analysis and compensation of transmission line tower data.

## 4. Reinforcement Learning Policy Simulation of Transmission Line Tower Data

Figure 2 shows the return value of each episode in the implementation process of A-Deep Q-Learning for transmission line tower data. They were used to check the usability of the proposed reinforcement learning. A total of 1000 transmission line tower data were applied to simulation, with the results measured through 5 repetitions. The Q-return value is obtained by adding the reward of Equation (4) obtained every time to each simulation (January 23–May 31, 2016). In Episode 1, the return value according to the deep Q-Learning policy was 54,148. When it was repeated over 50 episodes, the return value was 50,808. Policy exploration was possible to reduce the return values by as much as 6.1%. When about 50 episodes took place, the return value changed at the level of approximately 50,500~51,000. Here, the return values were changed in non-stationary form because there was a point where a weight of 1.2 was multiplied by Environment according to the excessive control of set temperature in the rewarding process as in Equations (16) and (17). Among the return values obtained from 1000 episodes, the biggest and smallest ones were 54,924 (10th episode) and 50,341 (46th episode), respectively. These results show the possibilities of realizing self-learning to explore and develop a policy for oneself through deep Q-Learning and learning and assessing outliers in transmission line tower data.

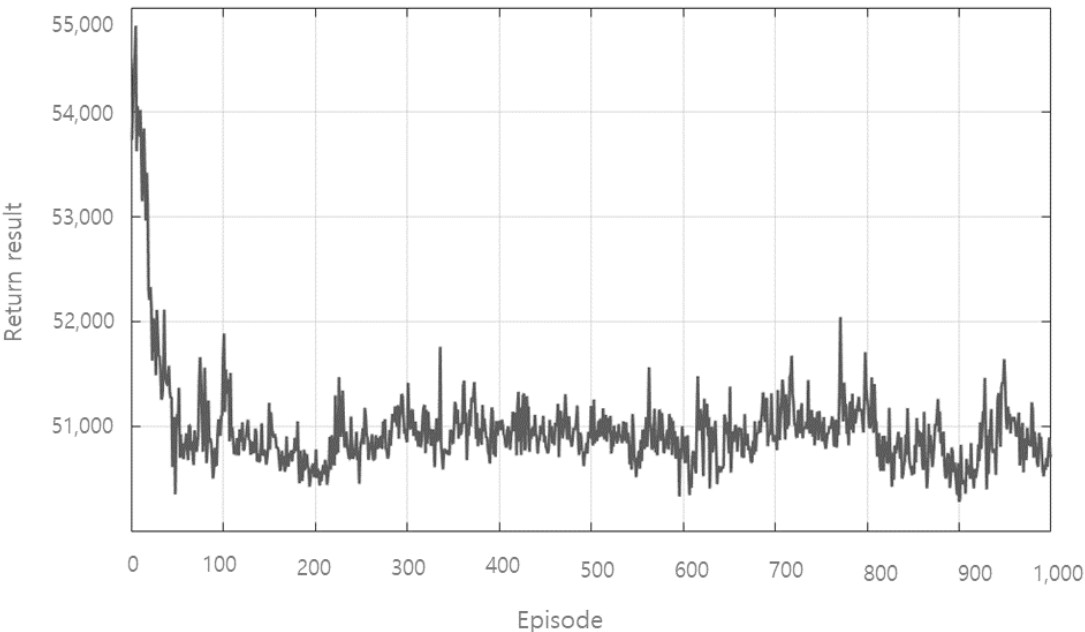

**Figure 2.** Return Value of Each Episode of Transmission-Line Tower Big Data.

## 5. Experiment

### 5.1. Big Data Set

This study sought to apply a learning model to identify outliers in transmission line tower data and check the connections between the identified data items. To do this, the following data were reflected in the performance evaluation: the data collected from transmission line tower sensors in Dalseong-gun, Daegu, Republic of Korea on January 23–May 31, 2016 and provided by Korea Electric Power Corporation. The collected data was 2.2 GB, containing 547,621 data (6,699,561 rows). Each of the rows was stored according to the data attributes including the transmission line tower number, location of equipment, temperature, humidity, pitch, roll, intensity of illumination, ultraviolet rays, pressure, remaining battery, and cycle. In addition, basic information including location, code, facility, date, time, pole, and position was collected and used in the analysis system. Among them, temperature, pitch, and roll data were extracted to analyze outliers in transmission line tower and power data and used in the performance evaluation. A total of 1.2 GB containing 250,251 data (3,383,241 rows) were identified and used in the experiment.

### 5.2. Clustering Result for the Outlier Recogniton of Transmission Line Tower Sensor Big Data

This study checked the altered K-means clustering results to find outliers in the transmission line tower data of transmission line towers. The proposed altered K-means chose principal components that were used to analyze multidimensional data for clustering and secure reference points to explain all the data as K the number of optimized clusters for choosing an automated K value. The outlier problem was solved by selecting data contained within 90% or higher in the normal distribution of candidate central points in the initial clusters and the central points of initial clusters according to Euclidean measurement. Clustering was performed for 250,251 pieces of transmission line tower data in certain areas in the experiment to check connections among K as the number of optimal clusters, size of outliers, and data. Data were distinguished through pre-treatment. Valid data to assess problems with transmission line towers were obtained from temperature, pitch, roll, and roll data. While temperature was used as data to check for problems with the body of a transmission line tower and the communication hull, pitch and roll served as data for checking the movements of transmission line towers. In this paper, the investigator conducted performance evaluation based on temperature, roll,

and pitch that could be used as response variables in the 22 sensor data sets collected from IoT sensors. The influence of a transmitting tower on a prediction model was taken into consideration as an element to enable the various applications of its external factors. More elements were selected based on the data whose frequency range was 20 or lower in the entire data scope of each item. While temperature represented the sensor value to measure the internal and external temperature of a transmitting tower, pitch and roll represented the measurement scope of acceleration sensors and were used to find a disorder with a transmitting tower by external force. Figure 3 shows the distribution of 250,251 data about temperature, pitch, and roll. Temperature was in the range of 9.7~36.2 degrees; pitch was in the range of 0~180 degrees, and roll was in the range of −85~0 degrees.

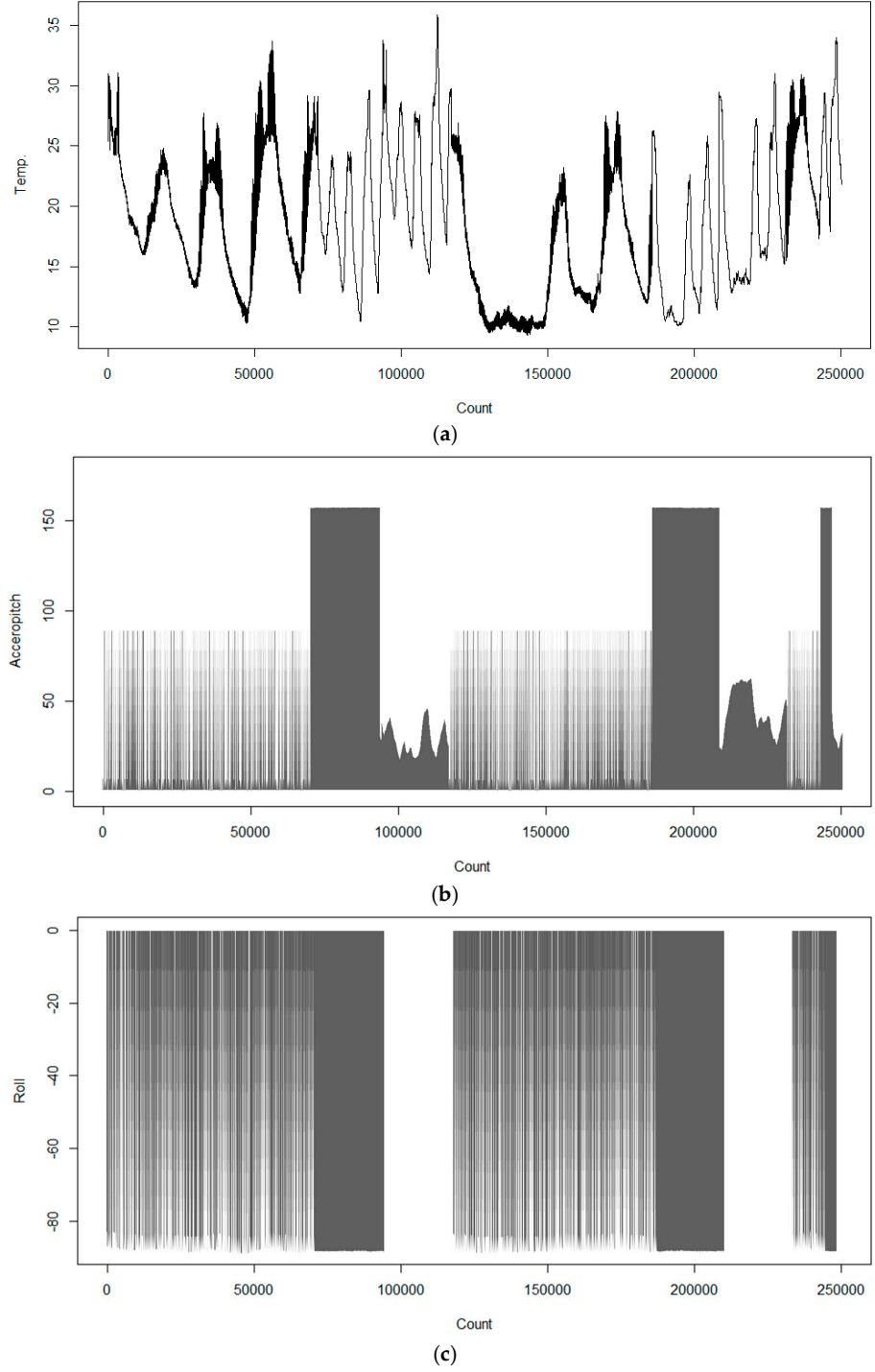

**Figure 3.** Scatter plot of electric pole data. (**a**) electric pole big data (temperature), (**b**) electric pole big data (acceropitch), (**c**) electric pole big data (roll).

When multidimensional information was reduced to four dimensions through principal component analysis, it explained 90% of the entire data set. As the number of optimal clusters, K was automatically set at 4 to proceed with each clustering. Figure 4a shows the results of clustering containing the pitch and roll data of transmission line towers, where a total of 465 outliers were identified. Figure 4b presents the results of clustering containing the pitch and temperature data of transmission line towers, where a total of 981 outliers were identified. Figure 4c illustrates the results of clustering containing the temperature and roll data of transmission line towers, where a total of 4485 outliers were identified.

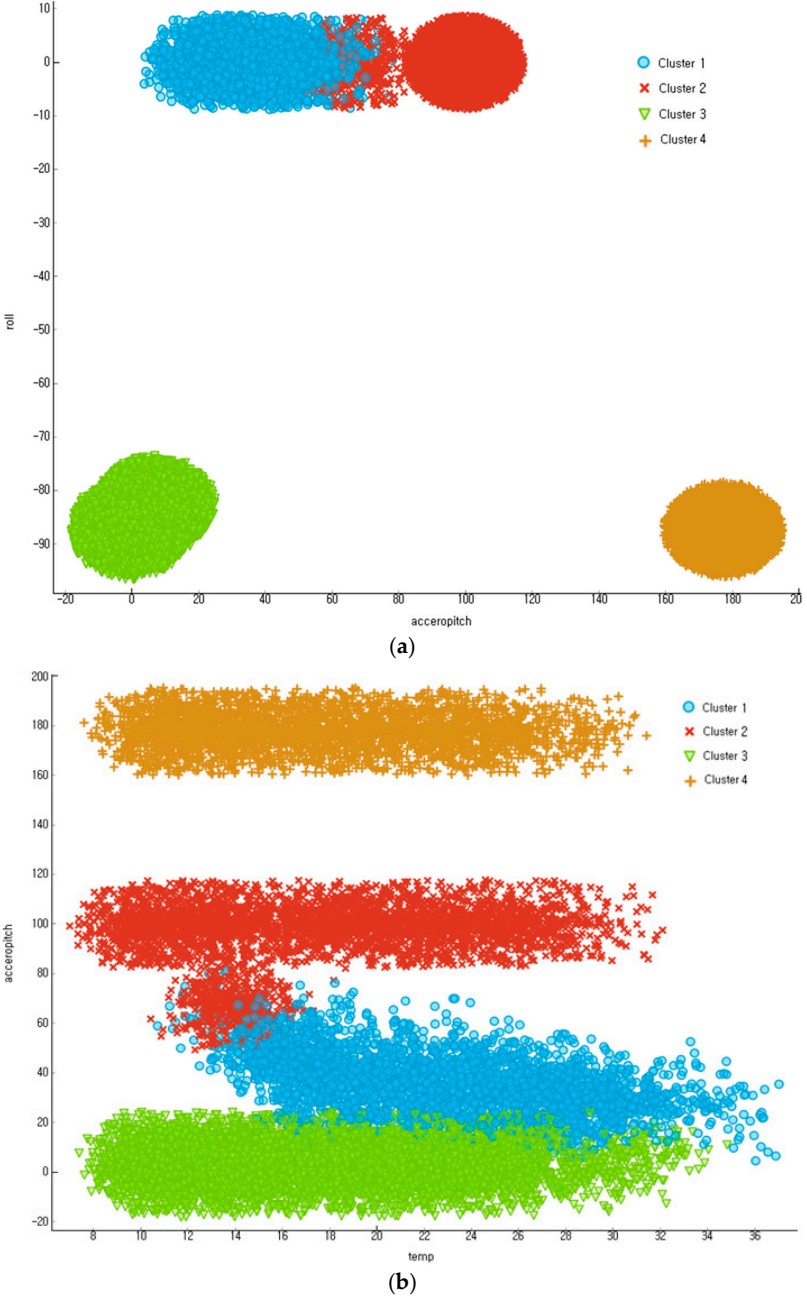

**Figure 4.** *Cont.*

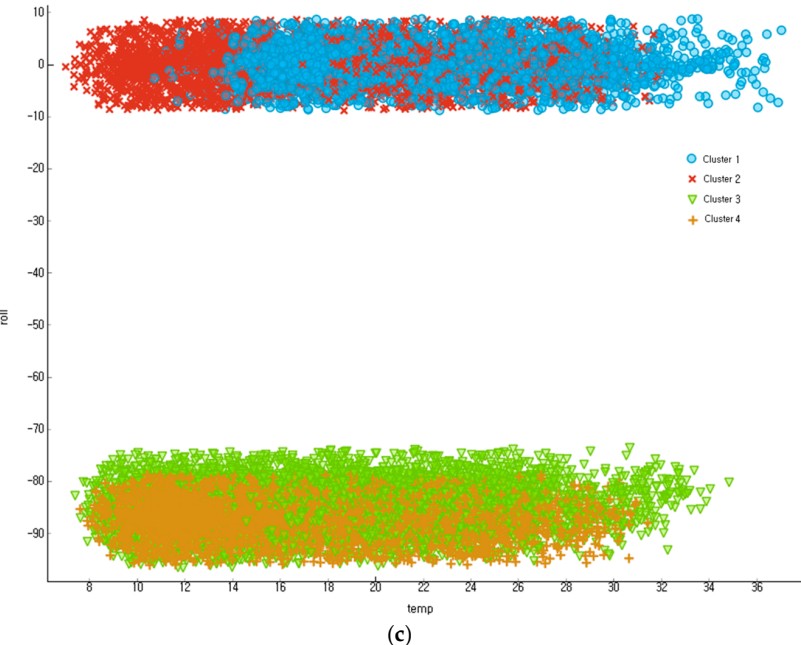

(**c**)

**Figure 4.** Result of clustering (K = 4) using Altered K-means in transmission line tower big data. (**a**) Clustering result in transmission line tower data (roll-acceropitch, K = 4); (**b**) clustering result in transmission line tower big data (acceropitch-temperature, K = 4). (**c**) Clustering result in transmission line tower big data (roll-temperature, K = 4).

This study assessed the sensitivity of outliers by approaching the initial central points of outliers for the analysis of clusters with a K-means algorithm. When the proposed algorithm was applied to the four places, the outlier data decreased in all of them. There were 465, 981, and 4485 outliers identified in temperature-pitch, pitch-temperature, and temperature-roll in Table 1. Since the initial central point of a cluster was designated at the outermost part based on the normal distribution of initial central points, outliers were identified in large numbers at the overlapping sections. The number of outliers was relatively small for the pitch and roll of transmission line towers because the changing values of pitch and roll moving up and down, right and left within a small scope were not recognized as a cluster. The number of outliers was relatively big when the temperature items were combined with clustering, since a new cluster was recognized in cases of changes to the main distribution sections day and night and abnormalities with transmission line towers.

**Table 1.** Clustering of Inlier-Outlier Detection (K = 4).

| Part | Cluster | lier | Data |
|---|---|---|---|
| Temp.-Pitch | 1 | Inlier | 63,414 |
| | | Outlier | 10 |
| | 2 | Inlier | 65,162 |
| | | Outlier | 464 |
| | 3 | Inlier | 61,138 |
| | | Outlier | 0 |
| | 4 | Inlier | 60,063 |
| | | Outlier | 0 |
| Pitch-Roll | 1 | Inlier | 62,987 |
| | | Outlier | 437 |
| | 2 | Inlier | 65,324 |
| | | Outlier | 302 |
| | 3 | Inlier | 60,896 |
| | | Outlier | 242 |
| | 4 | Inlier | 60,063 |
| | | Outlier | 0 |

**Table 1.** *Cont.*

| Part | Cluster | lier | Data |
|------|---------|------|------|
| Roll-Temp. | 1 | Inlier | 63,390 |
| | | Outlier | 34 |
| | 2 | Inlier | 62,942 |
| | | Outlier | 2,684 |
| | 3 | Inlier | 59,437 |
| | | Outlier | 1,701 |
| | 4 | Inlier | 59,997 |
| | | Outlier | 66 |

*5.3. ADQL Result for Outlier Learning*

The proposed algorithm was studied with the old DQN and LSTM_DQN A3C methods to demonstrate its efficacy. As for the condition to complete learning in the experiment, the proposed and old algorithms were studied when the reward function was measured as 0 or lower (when the prediction model assessed outlier data accurately with outlier prediction points) in the process of learning 5,981 out of 250,251 outlier data. Figure 5 shows the scope of learning completion according to a reward function over episodes. In the figure, the old DQN and A3C techniques enabled the earlier completion of learning because the goal was to predict outliers only within the scope of internal clustering containing outliers regardless of the clustering scope without continuous actions or natural actions (checking outliers). The reward feedback increased according to the number of input outliers, whose condition also slowed down the learning pace. This can disrupt fast learning at the moment, but high accuracy rates were recorded compared with outliers included within a certain scope of clusters. Figure 6 shows the prediction results of learning accuracy based on the clustering data (temperature-roll). A hundred predictions were made each time. The accuracy of outlier prediction points was assessed through 1,000 repetitions. The proposed algorithm recorded a prediction accuracy of 98.6% at repetition 800 and a higher prediction accuracy compared to previous studies at repetition 720. It posted about 1% lower prediction accuracy than the LSTM-DQN and DQN algorithms at Repetition 50 or lower, because all the data contained in the clustering scope were studied as outlier data. This resulted in the lower data accuracy of actual outliers. The algorithm increased its model predictability by about 2~3% at 720 times compared with the old LSTM-DQN, DQN, and A3C algorithms. In this study's performance evaluation, which was repeated 1,000 times at 70%, when the reinforcement learning of the prediction model was ended, as seen in Figure 6, the section marked the ending of outlier learning in the transmission line tower sensor data and the ending of the prediction model. These findings confirm that the proposed algorithm was superior to the old algorithms in the judgment of relearning and outliers.

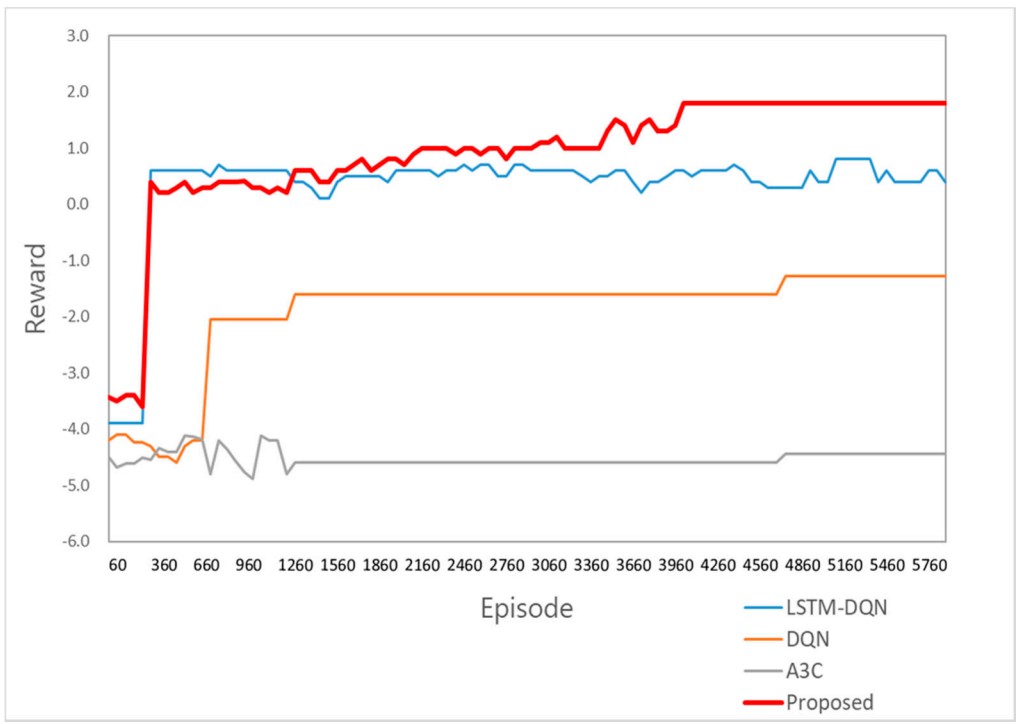

**Figure 5.** Scope of learning completion according to a reward function over episodes.

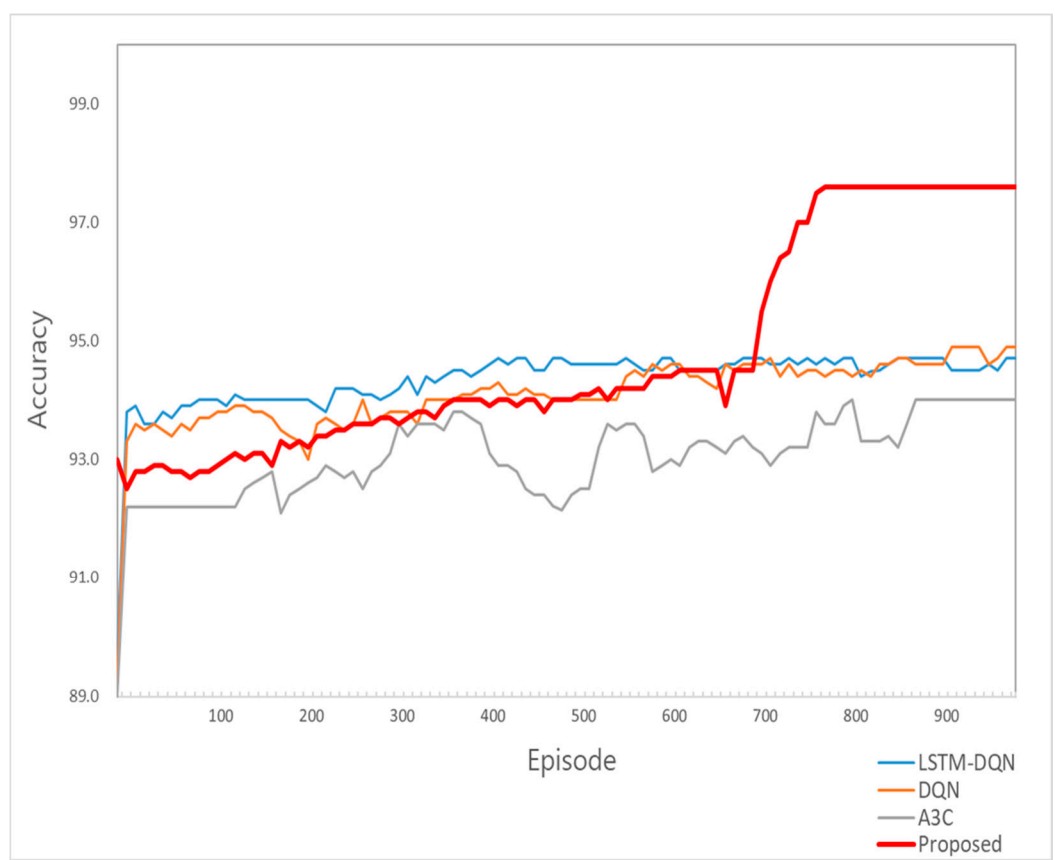

**Figure 6.** Prediction results of learning accuracy based on the clustering data.

## 5.4. Comparison of Transmission Line Tower Big Data Prediction System

Table 2 shows the direct comparison and evaluation results between the old prediction system for transmission line tower data and the proposed prediction system for transmission line tower data outliers. For the evaluation, each system was established with Python based on the algorithms proposed in each study. A total of 250,251 data of transmission line tower were entered in the performance evaluation. While 175,176 data accounting for 70% of the entire data set were studied in the learning model of transmission line tower outlier data, the remaining 30% or 75,076 data was applied to the test models. The performance evaluation results show that the proposed prediction model recorded a prediction rate of 94,688% (5,616) or approximately 2.29% ~ 4.19% higher than that of previous studies. The performance evaluation results show that the proposed prediction model recorded a prediction rate of 94,688% (5,616), which was approximately 2.29% ~ 4.19% higher than previous studies. The accuracy rate of the entire data model including outlier and normal data was 95.544% (236,987), which is approximately 0.8% ~ 4.3% higher than previous studies. These results show that the techniques of previous studies choosing the number of clusters randomly rather than automatically recorded lower prediction rates for the transmission line tower data that had new inputs of outlier data learning. Table 3 presents the learning time for 175,176 data and calculation time to measure 75,076 test data based on the model proposed in a previous study. The learning time for 175,176 data was reduced by approximately 6 ~ 20 seconds compared to the previous study. The test time was also reduced by approximately 3 ~ 8 seconds. The previous study reported that the time to measure test data was 200 seconds on average for data learning and model prediction. The analysis and calculation time of the algorithm proposed in the present study for the same power data prediction model was 182 seconds, down by about 18 seconds. The proposed algorithm improved the performance in terms of model prediction calculation time by 12.5% compared to the approximately 4.7% in the old study.

**Table 2.** Comparison of transmission line tower big data prediction system (prediction rate and accuracy rate).

| Part | Prediction System | Prediction Rate | Accuracy Rate |
|---|---|---|---|
| Study of [85] | K-means + Random Forest | 90.490% | 91.242% |
| Study of [86] | Hierarchical Clustering | 91.485% | 94.678% |
| Study of [87] | K-means+Sequence | 92.395% | 94.148% |
| Proposed Study | Altered K-means+ADQL | 94.688% | 95.544% |

**Table 3.** Comparison of transmission line tower data prediction system (computation time).

| Part | Prediction System | Learning Time(s) | Test Time(s) |
|---|---|---|---|
| Study of [85] | K-means + Random Forest | 157 | 34 |
| Study of [86] | Hierarchical Clustering | 164 | 39 |
| Study of [87] | K-means+Sequence | 171 | 37 |
| Proposed Study | Altered K-means+ADQL | 151 | 31 |

## 6. Conclusions

In recent years, energy-related industries have seen many research studies on smart transmission line tower big data analysis systems combining transmission line tower big data with information and communication technologies. Note, however, that these studies focus on the classification of transmission line tower big data patterns rather than transmission line tower itself; this makes it difficult to determine the semantic parts inside new data, which become important learning materials to determine the potential future outliers of transmission line tower. Thus, this study proposed a prediction model for future transmission line tower outlier data by combining an altered K-means algorithm with the A-Deep Q-Learning algorithm called reinforcement learning to learn transmission line tower outlier data for itself and determine meanings between the items of transmission line tower

big data. For this, this study proposed the following levels: the clustering level, which includes the transmission line tower data clustering technique containing automated cluster K values and transmission line tower outlier data; the reinforcement learning level, which checks the distance between data inside a cluster containing outliers with Q-value as a network output in the old DQN, and evaluates transmission line tower outlier data by applying the distance between internal data. The proposed research model recorded a prediction rate of 94.688% (5,616), which is approximately 2.29% ~ 4.19% higher than previous studies. It recorded an accuracy rate of 95.544% (236,987), which is approximately 0.8% ~ 4.3% higher than previous studies, for the entire data set containing outlier and normal data.

Future studies will investigate an improved analysis model for transmission line tower data in a data-deficient environment with only small amounts of transmission line tower data given rather than an environment for the analysis of large amounts of data by conducting research on a model to assess data through the reduction of scope of the reward function in the environment of reinforcement learning. This in turn is expected to increase the performance of the A-Deep Q-Learning proposed in this study.

**Author Contributions:** Conceptualization, S.-H.J. and J.-H.H.; Data curation, S.-H.J.; Formal analysis, S.-H.J. and J.-H.H.; Funding acquisition, S.-H.J.; Investigation, S.-H.J. and J.-H.H.; Methodology, S.-H.J.; Project administration, S.-H.J.; Resources, S.-H.J.; Software, S.-H.J.; Supervision, S.-H.J. and J.-H.H; Validation, J.-H.H; Visualization, S.-H.J. and J.-H.H.; Writing—original draft, S.-H.J. and J.-H.H.; Writing—review and editing, S.-H.J. and J.-H.H.

**Funding:** This work was supported by Youngsan University Research Fund of 2018. Also, this work was supported by the National Research Foundation of Korea (NRF) grant funded by the Korea government (MSIT) (NRF-2019R1G1A1002205).

**Conflicts of Interest:** The authors declare no conflict of interest.

# Appendix A

---

**Algorithm A1 Clustering Using PCA and Initial Centroid Subspace**

---

*Data: Non-labeling Transmission line tower Data Set*

*Output: Data by Cluster*

/\*The scatter plots and p as the number of data in the input data will be checked.\*/

Input:

      Training set $x^{(1)}$, $x^{(2)}$, $x^{(3)}$, ... $x^{(n)}$

           where $x^{(i)} \in \mathbb{R}^n$ (drop $x^1$=1 by convention)

/\* Principal component analysis will be conducted for all the input data entities, and principal components will be extracted up to the point where a constant value will be maintained to explain all the data. \*/

/\* The central point segmentation method will be applied to $C_{ki}$, number of random clusters $n_k$, and number of random central points based on the principal components extracted through principal component analysis. $m_k$ the central point of each initial cluster will be measured with a random cluster index vector. \*/

    repeat each $SK_i$, $\alpha \in \{1, 2, ..., n\}$, do

      if $SK_n > n$, which is each of all data $m_k$

        then

        for each $c_{ki} \in$ temporary centroid, do

          /\* $u_k$, $v_k$ represent the principal component direction and element \*/

            each data assigns $v_k = Y^T u_k / \lambda_k^{1/2}$

          update the temporary centroid number $n_k$

            assign the temporary centroid for scope($m_k$), $\overline{ny^2} - \lambda_1 < m_k < \overline{ny^2}$

        end

      end

/\* The minimum value of $m_k$, the central point of each segmented area, will be calculated with $A_k$ as the sum of squared distance to each entity. \*/

/\* $B_k$ as the minimum average distance between the central point of a random cluster and the entities included in an external cluster will be calculated.\*/

/\* S(k), which is the maximum cluster dissimilarity based on a difference between $A_k$ the degree of separation based on the average distance between the entities included in different clusters and all the other entities and $B_k$ as the degree of cohesion based on the average distance between an entity within a cluster and that in an external cluster will be treated as $C_k$, which represents K as the number of clusters.\*/

    for each temporary initial centroid($m_k$) $< SK_{i-1}$, do

      calculation from each data to cluster centroid(cohesion) $A_k$

          $A_k = \sum_{k=1}^{k} \sum_{i \in c_k} (X_i - m_k)^2$

      calculation from each data to cluster centroid (separation) $B_k$

      $B_k = \min\{(X_i - m_k)\}^2$

      assign initial centroid number S(k)

        $S(k) = \dfrac{1}{N} \dfrac{B_k - A_k}{\max(A_k, B_k)}$

    end

    if Selection -> Clustering number K

      then

      for check of each vector data, $\alpha \in c_k$

          each $c_k = \emptyset_{\mu,\sigma^2}(x_k, y_k) = \left(\dfrac{1}{\sigma\sqrt{2\pi}} e^{-\frac{(x-\mu)^2}{2\sigma^2}}\right) * \left(\dfrac{1}{\sigma\sqrt{2\pi}} e^{-\frac{(y-\mu)^2}{2\sigma^2}}\right)$

        if $P(X \geq x_k, y_k) = \pm 0.95 > c_k$, which is the initial centroid

          if there are two objects distributed

then assign the object to $m_1$ , the centroid of $C_1$, as the first cluster

else

then assign the object whose two vectors record the biggest length first to $m$, the centroid of $C_1$ as the first cluster

/* Measure the distance $(A_i)$ between the remaining objects $(x_i)$, the centroid of $C_1$ as the first cluster */

for check of each vector data, $\alpha \in c_i$

assign the object whose distance measurement is biggest to $m_2$ , the centroid of $C_2$ as the second cluster

$$C_2(m_2) = a_i \leftarrow \max_{1 \le i \le n}(A_k) = \max_{1 \le i \le n} \|x_i - m_1\| = \|a_i - m_1\|$$

for check of each vector data, $\alpha \in c_j$

assign the object with the maximum value to $m_3$, the centroid of $C_3$ as the third cluster

if (of the maximum measurements)

then　　$CA_i = \max \|x_j - m_1\|, \|x_j - m_1\|$
$$C_3(m_3) = x_i \leftarrow \max_{1 \le i \le n}(CA_i) = CA_1$$

until the remaining objects form a cluster in a direction closest to the area selected based on the initial values. The user will then appropriate the sum for the centroid of another cluster and repeat the Stages until there is no more travel of each cluster centroid.

　　　　　　　end for

　　　　end for

　end for main

---

**Algorithm A2** Transmission Line Tower Outlier Data Using Reinforcement Learning with Clustering as ADQL.

---

Data: Transmission line tower Clustering Data Set

Output: Outlier Data

Initialize experience memory L

Initialize parameters of representation Clustering Outlier $(\emptyset_R)$ and action scorer $(\emptyset_A)$ randomly

Initialize power clustering data value function with weight $Q^{Q'} \leftarrow \theta^Q$ neural network $Q'$

for episode = 1, A do

Initialize clustering and get start state description

for t=1, T do

then

if random () $< \epsilon$ do

Select random action $x_t$

else

Compute $Q_{(S_t, a)}$ for all actions using $\emptyset_A(V_s)$

execute

Action $a_t$, Observe reward $r_t$, Observe new state $s_{t+1}$

if $r_t > 0$

set priority $p_t = 1$

else

$p_t = 0$

Store $a_t$ to A

Store $(s_t, a_t, r_t, s_{t+1}, p_t)$ in D

Select random mini batch of transitions $(s_t, a_t, r_t, s_{t+1}, p_t)$ from D

With fraction $\gamma$ having $p_j = 1$

$$x_i = \begin{cases} \gamma_i & \text{if outlier is terminal} \\ \gamma_i + \gamma Q'\left(s_{i+1}, \mu'\left(s_{i+1}|\theta^{Q'}|\right)\right) & \text{if outlier is non-terminal} \end{cases}$$

Perform gradient descent step on loss $U = \frac{1}{N}\sum_i(y_i - Q(s_i, a_i|\theta^Q|))^2$ reduction and neural network update

if $A_t$ == inlier

then

learning outlier network $Q' \ne Q$

end for

end for main

---

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
