# Peer review of "A Novel on Transmission Line Tower Big Data Analysis Model Using Altered K-means and ADQL"

_sustainability, doi:10.3390/su11133499_

Reviewer 1 Report

This paper presents a model to analysis the Electric Power Data, both internal and external, it shows the advantages of the proposed model compare to the other exiting ones. The paper is well organized, the reviewer has the following comments:

1.       In the experiment part, the authors mainly compared the accuracy among all the methods. However, it would be better if the authors can take the performance/computation time into the consideration. If the proposed method takes much more computation power/time than other methods, then it might be a tradeoff.

2.       In figure 8, the accuracy of the proposed method increased dramatically after certain episode, the author can add some more discussion about that.

Overall, this paper shows a promising model which can predict the power data more accurately. But this needs to be based on the large amount of data which captured by large amount of sensors, which might need a large capital investment if need to be implemented in the real world. The authors can keep improving it reduce this kind of “need” in the future.

Author Response

Reply->

Dear Editor and Reviewers, respectfully,

I would first like to thank you for your comments, which are about performance and calculation time compared with previous studies. Since I did not reflect it during performance evaluation, I am particularly grateful for your comments. Based on what you pointed out, I added comparison and evaluation results with the old power prediction system. The time required to measure prediction results based on the same data and analysis model was added to "5.4 Performance evaluation with the old prediction system based on electric pole big data."

Add ) Table 3 shows learning time for 175,176 data and calculation time to measure 75,076 test data based on the model proposed in a previous study. The learning time for 175,176 data was reduced by approximately 6~20 seconds from the previous study. The test time was reduced by approximately 3~8 seconds. The previous study reported that the time to measure test data was 200 seconds on average for data learning and model prediction. The analysis and calculation time of the algorithm proposed in the present study for the same power data prediction model was 182 seconds down by about 18 seconds. The proposed algorithm improved performance for model prediction calculation time by 12.5% compared with about 4.7% in the old previous study.

Table 3. Comparison of electric pole data prediction system(Computation Time).

Part

Prediction System

Learning Time(s)

Test Time(s)

Study of [85]

K-means + Random Forest

157

34

Study of [86]

Hierarchical Clustering

164

39

Study of [87]

K-means+Sequence

171

37

Proposed Study

Altered K-means+ADQL

151

31

In figure 8, the accuracy of the proposed method increased dramatically after certain episode, the author can add some more discussion about that.

Reply-

I would first like to thank you for your comments. Your comments are about the part in which the accuracy rate increased when algorithm repetition recorded 720 times or more in Figure 8. I sorted out your comments as follows: the accuracy rate increased at 720 times or more when learning was ended at 70% or higher or measurement was completed for the learning scope of outlier as seen in Figure 7 in which feedback values were compensated by the reinforcement learning algorithm. The algorithm increased model prediction rates by about 2~3% compared with the old algorithm due to the rising accuracy rate under the relearning condition. The details were added to the body to reflect your comments.

Add) The algorithm increased its model predictability by about 2~3% at 720 times compared with the old LSTM-DQN, DQN, and A3C algorithms. In the study's performance evaluation that repeated 1,000 times at 70% when the reinforcement learning of the prediction model was ended as seen in Figure 5, the section marked the ending of outlier learning in the electric pole sensor data and the ending of the prediction model. These findings confirm that the proposed algorithm were superior to the old algorithms in the judgment of relearning and outlier.

Figure 5. Prediction results of learning accuracy based on the clustering data.

Overall, this paper shows a promising model which can predict the power data more accurately. But this needs to be based on the large amount of data which captured by large amount of sensors, which might need a large capital investment if need to be implemented in the real world. The authors can keep improving it reduce this kind of “need” in the future.

Reply-

I would first like to thank you for your comments. Your comments are about the stage in which data was secured from large amounts of sensor data. I completely agree with you on this. I invested a lot of time and money in securing data and classifying content of preprocessing in the study. It is especially difficult to access and secure national electric pole sensor data in large amounts at an individual research level rather than a joint project level. Researchers have pointed out these considerable difficulties. Trying to solve them, I plan to add research on the improvement of prediction performance in the data analysis model in case of data deficiency. This was addressed in the conclusion.

Add ) Future study will investigate an improved analysis model for electric power data in a data deficient environment with only small amounts of electric power data given rather than an environment for the analysis of large amounts of data by conducting research on a model to assess data by reducing the scope of reward function in the environment of reinforcement learning to increase the performance of A-Deep Q-Learning proposed in the present study.

(Extra Reply) The overall block diagram of the system proposed in the study was added to promote the understanding of the reviewers. Also added were related research data and related researches to propose a smart grid system through the analysis of electric pole sensor data. I respectfully request another chance at review based on these data and another round of detailed review.

Add 1) Figure 1 shows the block diagram of the system proposed in the study. The proposed analysis model for electric pole data consists of three levels: the row data level for electric pole data preprocessing including electric pole data collection and preprocessing, the clustering level to which an altered K-means algorithm would be applied, and the reinforcement learning level to learn how to check outlier in clustered electric pole sensor data for itself. The row data level involves gathering electric pole data, eliminating unnecessary data, and normalizing data. The clustering level includes an altered K-means algorithm to which principal component analysis proposed in the study will be applied. The reinforcement learning level includes a learning model to predict outlier in low level clustering to apply A-Deep Q-Learning altered from the off-policy and Q-table methods.

Add 2) Optimization at Reference

21.  J. Wu, K. Ota, M. Dong, J. Li and H. Wang, "Big Data Analysis-Based Security Situational Awareness for Smart Grid," in IEEE Transactions on Big Data, vol. 4, no. 3, pp. 408-417, 1 Sept., 2018.

22.  L. Sun, K. Zhou, X. Zhang and S. Yang, “Outlier Data Treatment Methods Toward Smart Grid Applications,” in IEEE Access, vol. 6, pp. 39849-39859, 2018.

23.  R. Menezes Salgado, T. Carvalho Machado and T. Ohishi, "Intelligent Models to Identification and Treatment of Outliers in Electrical Load Data," in IEEE Latin America Transactions, vol. 14, no. 10, pp. 4279-4286, Oct., 2016.

24.  D. Li, and S. K. Jayaweera, “Machine-learning aided optimal customer decision for an interactive smart grid,” IEEE Systems Journal, vol. 9, no. 4, pp.1529–1540, Dec., 2015.

25.  M. J. Ghorbani, M. A. Choudhry, and A. Feliachi, “A multiagent design for power distribution systems automation,” IEEE Transactions on Smart Grid, vol. 7, no. 1, pp. 329–339, Jan., 2016.

26.  Y. B. He, G. J. Mendis, and J. Wei, “Real-time detection of false data injection attacks in smart grid: a deep learning-based intelligent mechanism,” IEEE Transactions on Smart Grid, vol. 8, no. 5, pp. 2505–2516, Sep., 2017.

27.  G. K. Venayagamoorthy, R. K. Sharma, P. K. Gautam et al., “Dynamic energy management system for a smart microgrid,” IEEE Transactions on Neural Networks and Learning Systems, vol. 27, no.8, pp. 1643–1656, Aug., 2016.

28.  R. Thapa, L. Jiao, B. J. Oommen and A. Yazidi, "A Learning Automaton-Based Scheme for Scheduling Domestic Shiftable Loads in Smart Grids," in IEEE Access, vol. 6, pp. 5348-5361, 2018.

29.  P. Palensky, D. Dietrich, "Demand side management: Demand response intelligent energy systems and smart loads", IEEE Trans. Ind. Informat., vol. 7, no. 3, pp. 381-388, Aug., 2011.

69.  L. Yin, T. Yu, L. Zhou, L. Huang, X. Zhang, B. Zheng, "Artificial emotional reinforcement learning for automatic generation control of large-scale interconnected power grids", IET Gener. Transmiss. Distrib., vol. 11, no. 9, pp. 2305-2313, Jun., 2017.

70.  T. Yu, B. Zhou, K. W. Chan, L. Chen, B. Yang, "Stochastic optimal relaxed automatic generation control in non-markov environment based on multi-step Q(λ) learning", IEEE Trans. Power Syst., vol. 26, no. 3, pp. 1272-1282, Aug., 2011.

71.  Z. Yan and Y. Xu, "Data-Driven Load Frequency Control for Stochastic Power Systems: A Deep Reinforcement Learning Method With Continuous Action Search," in IEEE Transactions on Power Systems, vol. 34, no. 2, pp. 1653-1656, Mar., 2019.

72.  D. Zhang X. Han and C. Deng, “Review on the Research and Practice of Deep Learning and Reinforcement Learning in Smart Grids,” Journal of Power And Energy Systems, Vol. 4, No. 3, pp.362-370 Sep., 2018.

73.  F. Ruelens, B. J. Claessens, S. Vandeal et al., “Residential demand response of thermostatically controlled loads using batch reinforcement learning,” IEEE Transactions on Smart Grid, vol. 8, no.5, pp. 2149–2159, Sep., 2017.

74.  Z. Wen, D. O’ Neill, and H. Maei, “Optimal demand response using device-based reinforcement learning,” IEEE Transactions on Smart Grid, vol. 6, no. 5, pp. 2312–2324, Sep. 2015.

77.  Gang Ma, Linru Jiang, Guchao Xu, Jianyong Zheng, "A Model of Intelligent Fault Diagnosis of Power Equipment Based on CBR", Mathematical Problems in Engineering, vol. 2015, pp. 1, 2015.

78.  Connor Jennings, Dazhong Wu, Janis Terpenny, "Forecasting Obsolescence Risk and Product Life Cycle With Machine Learning", Components Packaging and Manufacturing Technology IEEE Transactions on, vol. 6, no. 9, pp. 1428-1439, 2016.

79.  P. Verma, P. Singh and R. D. S. Yadava, "Fuzzy c-means clustering based outlier detection for SAW electronic nose," 2017 2nd International Conference for Convergence in Technology (I2CT), Mumbai, 2017, pp. 513-519.

80.  M. Gupta, J. Gao, C.C. Aggarwal, J. Han, "Outlier detection for temporal data: A survey", IEEE Transactions on Knowledge and Data Engineering, vol. 26, pp. 2250-2267, 2014.

81.  W. Alves, D. Martins, U. Bezerra and A. Klautau, "A Hybrid Approach for Big Data Outlier Detection from Electric Power SCADA System," in IEEE Latin America Transactions, vol. 15, no. 1, pp. 57-64, Jan., 2017.

82.  J. Xiong et al., "Enhancing Privacy and Availability for Data Clustering in Intelligent Electrical Service of IoT," in IEEE Internet of Things Journal, vol. 6, no. 2, pp. 1530-1540, Apr., 2019.

83.  M. Salehi, C. Leckie, J. C. Bezdek, T. Vaithianathan and X. Zhang, "Fast Memory Efficient Local Outlier Detection in Data Streams," in IEEE Transactions on Knowledge and Data Engineering, vol. 28, no. 12, pp. 3246-3260, 1 Dec., 2016.

84.  R. Menezes Salgado, T. Carvalho Machado and T. Ohishi, "Intelligent Models to Identification and Treatment of Outliers in Electrical Load Data," in IEEE Latin America Transactions, vol. 14, no. 10, pp. 4279-4286, Oct., 2016.

We changed the title.

- (Before) A novel on Electric power Data Analysis Model using Altered K-means and ADQL

- (After) A novel on Electric Pole Big Data Analysis Model using Altered K-means and ADQL

Reviewer 2 Report

This study proposed a data analysis and prediction model for electric power outliers to assess something wrong with electric power data based on deep reinforcement learning. This is well written and can be accepted by this journal.

Author Response

This study proposed a data analysis and prediction model for electric power outliers to assess something wrong with electric power data based on deep reinforcement learning. This is well written and can be accepted by this journal.

Reply-

Dear Editor and Reviewers, respectfully,

The overall block diagram of the system proposed in the study was added to promote the understanding of the reviewers. Also added were related research data and related researches to propose a smart grid system through the analysis of electric pole sensor data. I respectfully request another chance at review based on these data and another round of detailed review.

Add 1) Figure 1 shows the block diagram of the system proposed in the study. The proposed analysis model for electric pole data consists of three levels: the row data level for electric pole data preprocessing including electric pole data collection and preprocessing, the clustering level to which an altered K-means algorithm would be applied, and the reinforcement learning level to learn how to check outlier in clustered electric pole sensor data for itself. The row data level involves gathering electric pole data, eliminating unnecessary data, and normalizing data. The clustering level includes an altered K-means algorithm to which principal component analysis proposed in the study will be applied. The reinforcement learning level includes a learning model to predict outlier in low level clustering to apply A-Deep Q-Learning altered from the off-policy and Q-table methods.

Add 2) Optimization at Reference

21.  J. Wu, K. Ota, M. Dong, J. Li and H. Wang, "Big Data Analysis-Based Security Situational Awareness for Smart Grid," in IEEE Transactions on Big Data, vol. 4, no. 3, pp. 408-417, 1 Sept., 2018.

22.  L. Sun, K. Zhou, X. Zhang and S. Yang, “Outlier Data Treatment Methods Toward Smart Grid Applications,” in IEEE Access, vol. 6, pp. 39849-39859, 2018.

23.  R. Menezes Salgado, T. Carvalho Machado and T. Ohishi, "Intelligent Models to Identification and Treatment of Outliers in Electrical Load Data," in IEEE Latin America Transactions, vol. 14, no. 10, pp. 4279-4286, Oct., 2016.

24.  D. Li, and S. K. Jayaweera, “Machine-learning aided optimal customer decision for an interactive smart grid,” IEEE Systems Journal, vol. 9, no. 4, pp.1529–1540, Dec., 2015.

25.  M. J. Ghorbani, M. A. Choudhry, and A. Feliachi, “A multiagent design for power distribution systems automation,” IEEE Transactions on Smart Grid, vol. 7, no. 1, pp. 329–339, Jan., 2016.

26.  Y. B. He, G. J. Mendis, and J. Wei, “Real-time detection of false data injection attacks in smart grid: a deep learning-based intelligent mechanism,” IEEE Transactions on Smart Grid, vol. 8, no. 5, pp. 2505–2516, Sep., 2017.

27.  G. K. Venayagamoorthy, R. K. Sharma, P. K. Gautam et al., “Dynamic energy management system for a smart microgrid,” IEEE Transactions on Neural Networks and Learning Systems, vol. 27, no.8, pp. 1643–1656, Aug., 2016.

28.  R. Thapa, L. Jiao, B. J. Oommen and A. Yazidi, "A Learning Automaton-Based Scheme for Scheduling Domestic Shiftable Loads in Smart Grids," in IEEE Access, vol. 6, pp. 5348-5361, 2018.

29.  P. Palensky, D. Dietrich, "Demand side management: Demand response intelligent energy systems and smart loads", IEEE Trans. Ind. Informat., vol. 7, no. 3, pp. 381-388, Aug., 2011.

69.  L. Yin, T. Yu, L. Zhou, L. Huang, X. Zhang, B. Zheng, "Artificial emotional reinforcement learning for automatic generation control of large-scale interconnected power grids", IET Gener. Transmiss. Distrib., vol. 11, no. 9, pp. 2305-2313, Jun., 2017.

70.  T. Yu, B. Zhou, K. W. Chan, L. Chen, B. Yang, "Stochastic optimal relaxed automatic generation control in non-markov environment based on multi-step Q(λ) learning", IEEE Trans. Power Syst., vol. 26, no. 3, pp. 1272-1282, Aug., 2011.

71.  Z. Yan and Y. Xu, "Data-Driven Load Frequency Control for Stochastic Power Systems: A Deep Reinforcement Learning Method With Continuous Action Search," in IEEE Transactions on Power Systems, vol. 34, no. 2, pp. 1653-1656, Mar., 2019.

72.  D. Zhang X. Han and C. Deng, “Review on the Research and Practice of Deep Learning and Reinforcement Learning in Smart Grids,” Journal of Power And Energy Systems, Vol. 4, No. 3, pp.362-370 Sep., 2018.

73.  F. Ruelens, B. J. Claessens, S. Vandeal et al., “Residential demand response of thermostatically controlled loads using batch reinforcement learning,” IEEE Transactions on Smart Grid, vol. 8, no.5, pp. 2149–2159, Sep., 2017.

74.  Z. Wen, D. O’ Neill, and H. Maei, “Optimal demand response using device-based reinforcement learning,” IEEE Transactions on Smart Grid, vol. 6, no. 5, pp. 2312–2324, Sep. 2015.

77.  Gang Ma, Linru Jiang, Guchao Xu, Jianyong Zheng, "A Model of Intelligent Fault Diagnosis of Power Equipment Based on CBR", Mathematical Problems in Engineering, vol. 2015, pp. 1, 2015.

78.  Connor Jennings, Dazhong Wu, Janis Terpenny, "Forecasting Obsolescence Risk and Product Life Cycle With Machine Learning", Components Packaging and Manufacturing Technology IEEE Transactions on, vol. 6, no. 9, pp. 1428-1439, 2016.

79.  P. Verma, P. Singh and R. D. S. Yadava, "Fuzzy c-means clustering based outlier detection for SAW electronic nose," 2017 2nd International Conference for Convergence in Technology (I2CT), Mumbai, 2017, pp. 513-519.

80.  M. Gupta, J. Gao, C.C. Aggarwal, J. Han, "Outlier detection for temporal data: A survey", IEEE Transactions on Knowledge and Data Engineering, vol. 26, pp. 2250-2267, 2014.

81.  W. Alves, D. Martins, U. Bezerra and A. Klautau, "A Hybrid Approach for Big Data Outlier Detection from Electric Power SCADA System," in IEEE Latin America Transactions, vol. 15, no. 1, pp. 57-64, Jan., 2017.

82.  J. Xiong et al., "Enhancing Privacy and Availability for Data Clustering in Intelligent Electrical Service of IoT," in IEEE Internet of Things Journal, vol. 6, no. 2, pp. 1530-1540, Apr., 2019.

83.  M. Salehi, C. Leckie, J. C. Bezdek, T. Vaithianathan and X. Zhang, "Fast Memory Efficient Local Outlier Detection in Data Streams," in IEEE Transactions on Knowledge and Data Engineering, vol. 28, no. 12, pp. 3246-3260, 1 Dec., 2016.

84.  R. Menezes Salgado, T. Carvalho Machado and T. Ohishi, "Intelligent Models to Identification and Treatment of Outliers in Electrical Load Data," in IEEE Latin America Transactions, vol. 14, no. 10, pp. 4279-4286, Oct., 2016.

We changed the title.

- (Before) A novel on Electric power Data Analysis Model using Altered K-means and ADQL

- (After) A Novel on Electric Pole Big Data Analysis Model Using Altered K-means and ADQL

Reviewer 3 Report

The paper proposes a machine-learning algorithm for data analysis of power system data applied under the concept of smart grids. 

1. Many parts of the paper can be presented in the appendix or omitted from the paper as the length of the paper is long.

2. Many parts of the paper such as the introduction are too wordy and in some cases not relevant to the main focus of the paper.

3. There are numerous examples of long run-on sentences and grammar errors (even in the title it seems that a word (approach) is missing). Editing by a professional English-speaking editor is recommended.

4. Some figures can be combined into on Figure with multiple subplots.

5. The terminology used in the paper is not very familiar for power system professionals and more matches with computer-science machine-learning experts. A review of this paper by power utility professionals is recommended.

6. legend need to be provided for all the figures.

7. All abbreviations need to be explained before used.

8. Too many keywords. Only keep the 5 most relevant keywords.

9. Pseudo-codes can be presented in the appendix. Comments are not really necessary.

10. More information on the data set need to be provided for the reader.

11. Verification using another data set is recommended as one experiment does not prove the validity and accuracy of the method.

12. There are many punctuation errors in the writing of the paper. Proper use of comma (,) is recommended.

13. Although, the literature review is complete but some papers from IEEE transactions and Journals which are very well related to this paper are missed such as (https://ieeexplore.ieee.org/stamp/stamp.jsp?arnumber=8404086) Disclaimer: the reviewer has no affiliation with this paper.

14. "electric power outlier" which is a term repeated in this paper numerously is an unfamiliar term and not very commonly-used. Revising this term is recommended.

15. In many cases the term "Power" used in this paper might confuses the reader with the concept of electric P = VI while this paper talks about data related to power or energy systems. The term "electric power data" need to be clarified and maybe changed. What data are we talking about? 

In equations presented in the paper, not all the parameters are introduced. Please revise.

Author Response

1. Many parts of the paper can be presented in the appendix or omitted from the paper as the length of the paper is long.

Reply-

Dear Editor and Reviewers, respectfully,

You have pointed out that I need to provide many parts of my paper in the Appendix or omit them. I totally agree with you on this. Writing this paper, I provided the content of engineering data analysis so that researchers both in the engineering and humanities field could check it. Combining your Comment No. 9 with this, I provided pseudocode in the Appendix. Thank you.

Add )  Algorithm 2 in the Appendix shows the altered K-means algorithm proposed in the study for the clustering of unlabeled data. Algorithm 4 in the Appendix shows the ADQL algorithm that is a reinforcement learning algorithm for the analysis and compensation of electric pole data.

Appendix

Algorithm 2 Clustering using PCA and Initial Centroid Subspace.

Data : Non-labeling Electric pole Data Set

Output : Data by Cluster

/*Scatter plots and p, the number of data, in the input data will be checked.*/

Input:

     Training set                                               , , , …

where (drop =1 by convention)

/* Principal component analysis will be conducted for all the input data entities, and principal components will be extracted until the point where a constant value will be maintained to explain all the data. */

/* The central point segmentation method will be applied to , the number of random clusters  and the number of random central points, based on the principal components that have been extracted through principal component analysis. , the central point of each initial cluster, will be measured with a random cluster index vector.  */

repeat each do

  if > n, which is each all data

     then

for each , do

   /* ,  represent the principal component direction and element */

      each data assign  =

update the temporary centroid number

      assign the temporary centroid for scope(),

end

end

/* The minimum value of , the central point of each segmented area, will be calculated with , the sum of squared distance to each entity. */

/*   will be calculated, which is the minimum average distance between the central point of a random cluster and the entities included in an external cluster.*/

/* S(k), which is the maximum cluster dissimilarity based on a difference between , the degree of separation based on the average distance between the entities included in different clusters and all the other entities, and , the degree of cohesion based on the average distance between an entity within a cluster and one in an external cluster, will be treated as , which represents the number of clusters, K.*/

for each temporary, do

calculation from each data to cluster centroid(cohesion),

calculation from each data to cluster centroid(separation),

assign the initial centroid number, S(k)

end

if Selection -> Clustering number K

then

for check of each vector data,

     each =

if > , which is the initial centroid

 if when there is two object distributed

then assign the object to  , the centroid of  , the first cluster

else

then assign the object whose two vectors record the biggest length first to , the centroid of   , the first cluster

/* Measure the distance() between the remaining objects() the centroid of  , the first cluster */

for check of each vector data,

asign the object whose distance measurement is the biggest to  , the centroid of  , the second cluster

for check of each vector data,

assign the object with the maximum value to , the centroid of , the third cluster

if (of the maximum measurements)

then  

until the remaining objects form a cluster in a direction that is the closest to the area selected based on the initial values. Then the user will appropriate the sum for the centroid of another cluster and repeat the Stages until there is no more travel of each cluster centroid.

end for

end for

end for main

Algorithm 4 Electric pole Outlier Data using Reinforcement learning with clustering as ADQL.

Data : Electric pole Clustering Data set

Output : Outlier Data

Initialize experience memory L

Initialize parameters of representation Clustering Outlier () and action scorer () randomly

Initialize power clustering data value function with weight  neural network

for episode=1,  do

Initialize clustering and get start state description

  for t=1, T do

     then

if random() << span=""> do

Select a random action

else

  Compute  for all actions using

execute

  Action , Observe reward , Observe new state

if  > 0

set priority = 1

    else

      = 0

   Store  to A

Store  in D

     Select random mini batch of transitions  from D

     With fraction  having = 1

Perform gradient descent step on loss  reduction and neural network update

    if  == inlier

        then

             learning outlier network  

end for

end for main

2. Many parts of the paper such as the introduction are too wordy and in some cases not relevant to the main focus of the paper.

Reply-

I appreciate that you found time to review my paper and give me comments for its revision despite your hectic schedule. Reflecting your comments, I connected the introduction to a need for an electric pole data analysis model(a recent social issue) and wrote it again. I also added a block diagram of the proposed system to the body in paper.

Add ) There has been an ongoing need for researches to analyze the data of outdoor electric poles in the domain of internal power IoT data analysis. electric poles form a power supply line along the streets and have recently caused several issues worldwide including spontaneous combustion and fire. In South Korea, a spontaneous combustion case happened in Gangneung-si in April, 2019 and caused big damage to human life and property. If there had been an analysis model for electric pole data, it would have helped to reduce the damage scope. Since electric poles are all equipped with sensors, it is easy to collect and analyze their data. They are, however, classified as power data managed by the government, which makes it difficult to secure the data. electric pole data consists of the followings :

3. There are numerous examples of long run-on sentences and grammar errors (even in the title it seems that a word (approach) is missing). Editing by a professional English-speaking editor is recommended.

Reply-

I appreciate that you found time to review my paper and give me comments for its revision despite your hectic schedule. The contents have been revised from the readers perspective with the assistance of a native English speaker and both the contribution. Thus, I’d like to respectfully request your re-review if possible. The contents added or changed are being highlighted in red.

4. Some figures can be combined into on Figure with multiple subplots.

Reply-

I appreciate that you found time to review my paper and give me comments for its revision despite your hectic schedule. As you have pointed out, I have added a figure(Figure 3~Figure 5 -> Figure 3). I once again request another review respectfully.

a. Clustering result in electric pole data (Roll-Acceropitch).

b. Clustering result in electric pole data(Acceropitch-Temperature).

c. Clustering result in electric pole data(Roll-Temperature).

Figure 3. Result of clustering (k=4) using altered K-means in electric pole data

5. The terminology used in the paper is not very familiar for power system professionals and more matches with computer-science machine-learning experts. A review of this paper by power utility professionals is recommended.

Reply-

I appreciate that you found time to review my paper and give me comments for its revision despite your hectic schedule. Trying to reflect your comments, I discussed the content again with an expert on power data analysis. As a joint project with the Korea Electric Power Corporation, this study aimed to analyze the data of electric poles along the streets and build a model to predict their outlier. I discussed the parts about data analysis again with the research team at the corporation. I thank you for your comments.

6. legend need to be provided for all the figures.

Reply-

I appreciate that you found time to review my paper and give me comments for its revision despite your hectic schedule. As you have pointed out, I added legends to all the figures in the paper. Thank you.

a. Clustering result in electric pole data (Roll-Acceropitch).

b. Clustering result in electric pole data(Acceropitch-Temperature).

c. Clustering result in electric pole data(Roll-Temperature).

Figure 3. Result of clustering (k=4) using altered K-means in electric pole data

Figure 4. Scope of learning completion according to a reward function over episodes.

Figure 5. Prediction results of learning accuracy based on the clustering data.

7. All abbreviations need to be explained before used.

Reply-

I appreciate that you found time to review my paper and give me comments for its revision despite your hectic schedule. As you have pointed out, I have added full names and explanations about them before their abbreviations.

8. Too many keywords. Only keep the 5 most relevant keywords.

Reply-

I appreciate that you found time to review my paper and give me comments for its revision despite your hectic schedule. As you have pointed out, I have narrowed down keywords to five that have the greatest relevance to the paper.

Keywords: Altered K-means; A-Deep Q Learning; Electric Pole Big Data Analysis; Big Data Analysis; Python.

9. Pseudo-codes can be presented in the appendix. Comments are not really necessary.

Reply-

I appreciate that you found time to review my paper and give me comments for its revision despite your hectic schedule. I provided pseudocode in the Appendix. Thank you.

Add )  Algorithm 2 in the Appendix shows the altered K-means algorithm proposed in the study for the clustering of unlabeled data. Algorithm 4 in the Appendix shows the ADQL algorithm that is a reinforcement learning algorithm for the analysis and compensation of electric pole data.

Appendix

Algorithm 2 Clustering using PCA and Initial Centroid Subspace.

Data : Non-labeling Electric pole Data Set

Output : Data by Cluster

/*Scatter plots and p, the number of data, in the input data will be checked.*/

Input:

     Training set , , , …

where (drop =1 by convention)

/* Principal component analysis will be conducted for all the input data entities, and principal components will be extracted until the point where a constant value will be maintained to explain all the data. */

/* The central point segmentation method will be applied to , the number of random clusters  and the number of random central points, based on the principal components that have been extracted through principal component analysis. , the central point of each initial cluster, will be measured with a random cluster index vector.  */

repeat each do

  if > n, which is each all data

     then

for each , do

   /* ,  represent the principal component direction and element */

      each data assign  =

update the temporary centroid number

      assign the temporary centroid for scope(),

end

end

/* The minimum value of , the central point of each segmented area, will be calculated with , the sum of squared distance to each entity. */

/*   will be calculated, which is the minimum average distance between the central point of a random cluster and the entities included in an external cluster.*/

/* S(k), which is the maximum cluster dissimilarity based on a difference between , the degree of separation based on the average distance between the entities included in different clusters and all the other entities, and , the degree of cohesion based on the average distance between an entity within a cluster and one in an external cluster, will be treated as , which represents the number of clusters, K.*/

for each temporary, do

calculation from each data to cluster centroid(cohesion),

calculation from each data to cluster centroid(separation),

assign the initial centroid number, S(k)

end

if Selection -> Clustering number K

then

for check of each vector data,

     each =

if > , which is the initial centroid

 if when there is two object distributed

then assign the object to  , the centroid of  , the first cluster

else

then assign the object whose two vectors record the biggest length first to , the centroid of   , the first cluster

/* Measure the distance() between the remaining objects() the centroid of  , the first cluster */

for check of each vector data,

asign the object whose distance measurement is the biggest to  , the centroid of  , the second cluster

for check of each vector data,

assign the object with the maximum value to , the centroid of , the third cluster

if (of the maximum measurements)

then  

until the remaining objects form a cluster in a direction that is the closest to the area selected based on the initial values. Then the user will appropriate the sum for the centroid of another cluster and repeat the Stages until there is no more travel of each cluster centroid.

end for

end for

end for main

Algorithm 4 Electric pole Outlier Data using Reinforcement learning with clustering as ADQL.

Data : Electric pole Clustering Data set

Output : Outlier Data

Initialize experience memory L

Initialize parameters of representation Clustering Outlier () and action scorer () randomly

Initialize power clustering data value function with weight  neural network

for episode=1,  do

Initialize clustering and get start state description

  for t=1, T do

     then

if random() << span=""> do

Select a random action

else

  Compute  for all actions using

execute

  Action , Observe reward , Observe new state

if  > 0

set priority = 1

    else

      = 0

   Store  to A

Store  in D

     Select random mini batch of transitions  from D

     With fraction  having = 1

Perform gradient descent step on loss  reduction and neural network update

    if  == inlier

        then

             learning outlier network  

end for

end for main

10. More information on the data set need to be provided for the reader.

Reply-

I appreciate that you found time to review my paper and give me comments for its revision despite your hectic schedule. As you have pointed out, I have added explanations about the data sets in the study as follows:

Add )  The following data was reflected in performance evaluation to check this: the data was collected from electric pole sensors in Dalseong-gun, Daegu, South Korea on January 23~May 31, 2016 and provided by the Korea Electric Power Corporation. The collected data was 2.2GB, containing 547,621 data(6,699,561 rows). Each of the rows was stored by the data attributes including the electric pole number, location of equipment, temperature, humidity, pitch, roll, intensity of illumination, ultraviolet rays, pressure, remaining battery, and cycle. In addition, basic information including location, code, facility, date, time, pole, and position was collected and used in the analysis system. Of them, temperature, pitch, and roll data was extracted to analyze outliers in electric pole and power data and used in performance evaluation. A total of 1.2GB, containing 250,251 data(3,383,241 rows) were identified and used in the experiment.

11. Verification using another data set is recommended as one experiment does not prove the validity and accuracy of the method.

Reply-

I appreciate that you found time to review my paper and give me comments for its revision despite your hectic schedule. Reflecting your comments, I have added content about the effectiveness and accuracy of the data provided in the study. Tables 2 and 3 offer detailed explanations about the data, reasons for the higher accuracy of the analysis model than previous studies, and improvement in performance time compared with other researchers. I once again request another review respectfully.

Add ) Table 2 shows the direct comparison and evaluation results between the old prediction system for electric pole data and the proposed prediction system for electric pole data outliers. For evaluation, each system was established with Python based on the algorithms proposed in each study. A total of 250,251 data of electric pole were entered in performance evaluation. While 175,176 data accounting for 70% of the entire data set were studied in the learning model of electric pole outlier data, the remaining 30% or 75,076 data were applied to the test models. The performance evaluation results show that the proposed prediction model recorded a prediction rate of 94,688% (5,616), which was approximately 2.29%~4.19% higher than previous studies. The accuracy rate of the entire data model including outlier and normal data was 95.544% (236,987), which is approximately 0.8% ~ 4.3% higher than previous studies. These results show that the techniques of previous studies choosing the number of clusters randomly rather than automatically recorded lower prediction rates for the electric pole data that had new inputs of outlier data learning. Table 3 shows learning time for 175,176 data and calculation time to measure 75,076 test data based on the model proposed in a previous study. The learning time for 175,176 data was reduced by approximately 6~20 seconds from the previous study. The test time was reduced by approximately 3~8 seconds. The previous study reported that the time to measure test data was 200 seconds on average for data learning and model prediction. The analysis and calculation time of the algorithm proposed in the present study for the same power data prediction model was 182 seconds down by about 18 seconds. The proposed algorithm improved performance for model prediction calculation time by 12.5% compared with about 4.7% in the old previous study.

Table 2. Comparison of electric pole data prediction system(Prediction Rate and Accuracy Rate).

Part

Prediction System

Prediction Rate

Accuracy Rate

Study of [85]

K-means + Random Forest

90.490%

91.242%

Study of [86]

Hierarchical Clustering

91.485%

94.678%

Study of [87]

K-means+Sequence

92.395%

94.148%

Proposed Study

Altered K-means+ADQL

94.688%

95.544%

Table 3. Comparison of electric pole data prediction system(Computation Time).

Part

Prediction System

Learning Time(s)

Test Time(s)

Study of [85]

K-means + Random Forest

157

34

Study of [86]

Hierarchical Clustering

164

39

Study of [87]

K-means+Sequence

171

37

Proposed Study

Altered K-means+ADQL

151

31

12. There are many punctuation errors in the writing of the paper. Proper use of comma (,) is recommended.

Reply-

I appreciate that you found time to review my paper and give me comments for its revision despite your hectic schedule. As you have pointed out, I have corrected my punctuation errors except for the parts with some terms. Thank you.

13. Although, the literature review is complete but some papers from IEEE transactions and Journals which are very well related to this paper are missed such as (https://ieeexplore.ieee.org/stamp/stamp.jsp?arnumber=8404086) Disclaimer: the reviewer has no affiliation with this paper.

Reply-

I appreciate that you found time to review my paper and give me comments for its revision despite your hectic schedule. As you have pointed out, I have added researches on electric pole sensor data analysis, reinforcement learning, and smart grid related to my paper. Thank you.

Add 1) Optimization at Reference

21.  J. Wu, K. Ota, M. Dong, J. Li and H. Wang, "Big Data Analysis-Based Security Situational Awareness for Smart Grid," in IEEE Transactions on Big Data, vol. 4, no. 3, pp. 408-417, 1 Sept., 2018.

22.  L. Sun, K. Zhou, X. Zhang and S. Yang, “Outlier Data Treatment Methods Toward Smart Grid Applications,” in IEEE Access, vol. 6, pp. 39849-39859, 2018.

23.  R. Menezes Salgado, T. Carvalho Machado and T. Ohishi, "Intelligent Models to Identification and Treatment of Outliers in Electrical Load Data," in IEEE Latin America Transactions, vol. 14, no. 10, pp. 4279-4286, Oct., 2016.

24.  D. Li, and S. K. Jayaweera, “Machine-learning aided optimal customer decision for an interactive smart grid,” IEEE Systems Journal, vol. 9, no. 4, pp.1529–1540, Dec., 2015.

25.  M. J. Ghorbani, M. A. Choudhry, and A. Feliachi, “A multiagent design for power distribution systems automation,” IEEE Transactions on Smart Grid, vol. 7, no. 1, pp. 329–339, Jan., 2016.

26.  Y. B. He, G. J. Mendis, and J. Wei, “Real-time detection of false data injection attacks in smart grid: a deep learning-based intelligent mechanism,” IEEE Transactions on Smart Grid, vol. 8, no. 5, pp. 2505–2516, Sep., 2017.

27.  G. K. Venayagamoorthy, R. K. Sharma, P. K. Gautam et al., “Dynamic energy management system for a smart microgrid,” IEEE Transactions on Neural Networks and Learning Systems, vol. 27, no.8, pp. 1643–1656, Aug., 2016.

28.  R. Thapa, L. Jiao, B. J. Oommen and A. Yazidi, "A Learning Automaton-Based Scheme for Scheduling Domestic Shiftable Loads in Smart Grids," in IEEE Access, vol. 6, pp. 5348-5361, 2018.

29.  P. Palensky, D. Dietrich, "Demand side management: Demand response intelligent energy systems and smart loads", IEEE Trans. Ind. Informat., vol. 7, no. 3, pp. 381-388, Aug., 2011.

69.  L. Yin, T. Yu, L. Zhou, L. Huang, X. Zhang, B. Zheng, "Artificial emotional reinforcement learning for automatic generation control of large-scale interconnected power grids", IET Gener. Transmiss. Distrib., vol. 11, no. 9, pp. 2305-2313, Jun., 2017.

70.  T. Yu, B. Zhou, K. W. Chan, L. Chen, B. Yang, "Stochastic optimal relaxed automatic generation control in non-markov environment based on multi-step Q(λ) learning", IEEE Trans. Power Syst., vol. 26, no. 3, pp. 1272-1282, Aug., 2011.

71.  Z. Yan and Y. Xu, "Data-Driven Load Frequency Control for Stochastic Power Systems: A Deep Reinforcement Learning Method With Continuous Action Search," in IEEE Transactions on Power Systems, vol. 34, no. 2, pp. 1653-1656, Mar., 2019.

72.  D. Zhang X. Han and C. Deng, “Review on the Research and Practice of Deep Learning and Reinforcement Learning in Smart Grids,” Journal of Power And Energy Systems, Vol. 4, No. 3, pp.362-370 Sep., 2018.

73.  F. Ruelens, B. J. Claessens, S. Vandeal et al., “Residential demand response of thermostatically controlled loads using batch reinforcement learning,” IEEE Transactions on Smart Grid, vol. 8, no.5, pp. 2149–2159, Sep., 2017.

74.  Z. Wen, D. O’ Neill, and H. Maei, “Optimal demand response using device-based reinforcement learning,” IEEE Transactions on Smart Grid, vol. 6, no. 5, pp. 2312–2324, Sep. 2015.

77.  Gang Ma, Linru Jiang, Guchao Xu, Jianyong Zheng, "A Model of Intelligent Fault Diagnosis of Power Equipment Based on CBR", Mathematical Problems in Engineering, vol. 2015, pp. 1, 2015.

78.  Connor Jennings, Dazhong Wu, Janis Terpenny, "Forecasting Obsolescence Risk and Product Life Cycle With Machine Learning", Components Packaging and Manufacturing Technology IEEE Transactions on, vol. 6, no. 9, pp. 1428-1439, 2016.

79.  P. Verma, P. Singh and R. D. S. Yadava, "Fuzzy c-means clustering based outlier detection for SAW electronic nose," 2017 2nd International Conference for Convergence in Technology (I2CT), Mumbai, 2017, pp. 513-519.

80.  M. Gupta, J. Gao, C.C. Aggarwal, J. Han, "Outlier detection for temporal data: A survey", IEEE Transactions on Knowledge and Data Engineering, vol. 26, pp. 2250-2267, 2014.

81.  W. Alves, D. Martins, U. Bezerra and A. Klautau, "A Hybrid Approach for Big Data Outlier Detection from Electric Power SCADA System," in IEEE Latin America Transactions, vol. 15, no. 1, pp. 57-64, Jan., 2017.

82.  J. Xiong et al., "Enhancing Privacy and Availability for Data Clustering in Intelligent Electrical Service of IoT," in IEEE Internet of Things Journal, vol. 6, no. 2, pp. 1530-1540, Apr., 2019.

83.  M. Salehi, C. Leckie, J. C. Bezdek, T. Vaithianathan and X. Zhang, "Fast Memory Efficient Local Outlier Detection in Data Streams," in IEEE Transactions on Knowledge and Data Engineering, vol. 28, no. 12, pp. 3246-3260, 1 Dec., 2016.

84.  R. Menezes Salgado, T. Carvalho Machado and T. Ohishi, "Intelligent Models to Identification and Treatment of Outliers in Electrical Load Data," in IEEE Latin America Transactions, vol. 14, no. 10, pp. 4279-4286, Oct., 2016.

14. "electric power outlier" which is a term repeated in this paper numerously is an unfamiliar term and not very commonly-used. Revising this term is recommended.

Reply-

I appreciate that you found time to review my paper and give me comments for its revision despite your hectic schedule. Reflecting your comments, I have replaced the term, electric power outlier, with electric pole sensor data and revised the entire paper and its title accordingly. Thank you.

(ex.) We changed the title.

- (Before) A novel on Electric power Data Analysis Model using Altered K-means and ADQL

- (After) A novel on Electric Pole Big Data Analysis Model using Altered K-means and ADQL

15. In many cases the term "Power" used in this paper might confuses the reader with the concept of electric P = VI while this paper talks about data related to power or energy systems. The term "electric power data" need to be clarified and maybe changed. What data are we talking about?

Reply-

I appreciate that you found time to review my paper and give me comments for its revision despite your hectic schedule. Your comments are about my mistake that made the reviewers confused about the power data. Reflecting your comments, I have replaced power data with electric pole sensor data as seen in Comment No. 14. I once again request another review respectfully.

Add ) There are various researches being conducted on a prediction system for power-related data[77-82]. In recent years, active research efforts have been made to investigate pattern mining such as power demand and patterns and analyze outlier to identify defective data in the collected data[83-84]. The present study compared and evaluated the model proposed in it with three researches below to analyze electric pole sensor data of power-related data[85-87].

Round  2

Reviewer 3 Report

Thank you for all the modifications implemented in this revision. 

The term power pole is not appropriate (as it is mainly used in the context of poles and zeros of a system). I recommend using the term "transmission-line tower" or "Transmission Tower".

The title of the paper can be modified accordingly:  "A Novel Model on Transmission-Line Tower Big-Data Analysis Using Altered K-means and ADQL"

Can Figure 3. parts b and c be presented with more explanation on the black boxes? Can you zoom on those sections and provide a snapshot?

Would it be possible to add grid lines for the figures (similar to figure 2)?

Formatting of the references in terms of lower case capital letters and italic formatting for the journal/conference titles is not consistent and standard. It can be modified further.

The paper look much better with all the added modifications. I hope you can apply the suggested further minor suggestions. 

Thank you

Author Response

The term power pole is not appropriate (as it is mainly used in the context of poles and zeros of a system). I recommend using the term "transmission-line tower" or "Transmission Tower". The title of the paper can be modified accordingly: "A Novel Model on Transmission-Line Tower Big-Data Analysis Using Altered K-means and ADQL"

Reply-

I appreciate that you found time to review my paper and give me comments for its revision despite your hectic schedule. Reflecting your comments, I have replaced the term, electric pole, with transmission line tower and revised the entire paper and its title accordingly. Thank you.

(ex.) We changed the title.

- (Before) A novel on Electric Pole Big Data Analysis Model using Altered K-means and ADQL

- (After) A novel on Transmission Line Tower Big Data Analysis Model using Altered K-means and ADQL

Can Figure 3. parts b and c be presented with more explanation on the black boxes? Can you zoom on those sections and provide a snapshot? Would it be possible to add grid lines for the figures (similar to figure 2)?

Reply-

I appreciate that you found time to review my paper and give me comments for its revision despite your hectic schedule. Reflecting your comments, I have added explanations about data based on your comments. As for what you pointed out regarding the output line in the graph, I have to tell you that the output of a grid line like the one in Figure 2 is impossible due to the big deviation of data in pitch and roll. Revision were, however, made to the graph to check data distribution.

Add)  In this paper, the investigator conducted performance evaluation with temperature, roll, and pitch that could be used as response variables in the 22 sensor data sets collected from IoT sensors. The influence of a transmitting tower on a prediction model was taken into consideration as an element to enable the various applications of its external factors. More elements were selected based on the data whose frequency range was 20 or lower in the entire data scope of each item. While temperature represented the sensor value to measure the internal and external temperature of a transmitting tower, pitch and roll represented the measurement scope of acceleration sensors and were used to find a disorder with a transmitting tower by external force. Figure 3 shows the distribution of 250,251 data about temperature, pitch, and roll. Temperature was in the range of 9.7 ~ 36.2 degrees; pitch was in the range of 0 ~ 180 degrees, and roll, in the range of -85 ~ 0 degrees.

a. Electric Pole Data (Temperature)

b. Electric Pole Data (Acceropitch)

c. Electric Pole Data (Roll)

Figure 3 Scatter Plot of Electric Pole Data.

Formatting of the references in terms of lower case capital letters and italic formatting for the journal/conference titles is not consistent and standard. It can be modified further.

Reply-

I appreciate that you found time to review my paper and give me comments for its revision despite your hectic schedule. Reflecting your comments, I have changed my paper in reference part.

References

1.     J. Q. Trelewicz, “Big data and big money: The role of data in the financial sector,” IT Professional, vol. 19, no. 3, pp. 8–10, 2017.

2.    E. I. Lab, “Big data in banking for marketers how to derive value from big data, ”White Paper.

3.    U. Srinivasan and B. Arunasalam, “Leveraging big data analytics to reduce healthcare costs,” IT professional, vol. 15, no. 6, pp. 21–28, 2013.

4.    M. M. Islam, M. A. Razzaque, M. M. Hassan,W. N. Ismail, and B. Song, “Mobile cloud-based big healthcare data processing in smart cities,” IEEE Access, vol. 5, pp. 11, 887–11 899, 2017.

5.    M. Marjani, F. Nasaruddin, A. Gani, A. Karim, I. A. T. Hashem, A. Siddiqa, and I. Yaqoob, “Big IoT data analytics: Architecture, opportunities, and open research challenges,” IEEE Access, vol. 5, pp. 5247–5261, 2017.

6.     M. Satyanarayanan, P. Simoens, Y. Xiao, P. Pillai, Z. Chen, K. Ha, W. Hu, and B. Amos, “Edge analytics in the internet of things,” IEEE Pervasive Computing, vol. 14, no. 2, pp. 24–31, 2015.

7.     S. K. Sharma and X.Wang, “Live data analytics with collaborative edge and cloud processing in wireless iot networks,” IEEE Access, vol. 5, pp. 4621–4635, 2017.

8.     X. He, Q. Ai, R. C. Qiu, W. Huang, L. Piao, and H. Liu, “A big data architecture design for smart grids based on random matrix theory,” IEEE Transactions on Smart Grid, vol. 8, no. 2, pp. 674–686, March 2017.

9.     Y. Sun, H. Song, A. J. Jara, and R. Bie, “Internet of things and big data analytics for smart and connected communities,” IEEE Access, vol. 4, pp. 766–773, 2016.

10.    P. Ta-Shma, A. Akbar, G. Gerson-Golan, G. Hadash, F. Carrez, and K. Moessner,“An ingestion and analytics architecture for iot applied to smart city use cases,” IEEE Internet of Things Journal, 2017.

11.    K. Wedgwood and R. Howard, “Big data and analytics in travel and transportation,” IBM Big Data and Analytics White Paper, 2014.

12.   T. Hong, “Data analytics: Making the smart grid smarter [guest editorial],” IEEE Power and Energy Magazine, vol. 16, no. 3, pp. 12–16, May 2018.

13.    A. Bose, “Smart transmission grid applications and their supporting infrastructure,” IEEE Transactions on Smart Grid, vol. 1, no. 1, pp. 11–19, June, 2010.

14.    A. P. S. Meliopoulos, G. Cokkinides, R. Huang, E. Farantatos, S. Choi, Y. Lee, and X. Yu, “Smart grid technologies for autonomous operation and control,” IEEE Transactions on Smart Grid, vol. 2, no. 1, pp. 1–10, March, 2011.

15.    G. T. Heydt, “The next generation of power distribution systems,” IEEE Transactions on Smart Grid, vol. 1, no. 3, pp. 225–235, Dec 2010.

16.    W. Hou, Z. Ning, L. Guo, and X. Zhang, “Temporal, functional and spatial big data computing framework for large-scale smart grid,” IEEE Transactions on Emerging Topics in Computing, pp. 1–1, 2018.

17.    K. Zhou, C. Fu, and S. Yang, “Big data driven smart energy management: From big data to big insights,” Renewable and Sustainable Energy Reviews, vol. 56, pp. 215–225, 2016.

18.    Bhattarai, B. P., Paudyal, S., Luo, Y., Mohanpurkar, M., Cheung, K., Tonkoski, R.,  & Manic, M.  Big data analytics in smart grids: state-of-the-art, challenges, opportunities, and future directions. IET Smart Grid., pp.1-15, 2019.

19.    Seung-Mo Je, Jun-Ho Huh.; "An Optimized Algorithm and Test Bed for Improvement of Efficiency of ESS and Energy Use." Electronics, MDPI, Vol.7, No.12, pp.1-25, 2018.

20.    Seung-Mo Je, Jun-Ho Huh.; "Estimation of Future Power Consumption Level in Smart Grid: Application of Fuzzy Logic and Genetic Algorithm on Big Data Platform", International Journal of Communication Systems, Wiley, Accepted, 2019.

21.  J. Wu, K. Ota, M. Dong, J. Li and H. Wang, "Big Data Analysis-Based Security Situational Awareness for Smart Grid," IEEE Transactions on Big Data, vol. 4, no. 3, pp. 408-417, 1 Sept., 2018.

22.  L. Sun, K. Zhou, X. Zhang and S. Yang, “Outlier Data Treatment Methods Toward Smart Grid Applications,” IEEE Access, vol. 6, pp. 39849-39859, 2018.

23.  R. Menezes Salgado, T. Carvalho Machado and T. Ohishi, "Intelligent Models to Identification and Treatment of Outliers in Electrical Load Data," IEEE Latin America Transactions, vol. 14, no. 10, pp. 4279-4286, Oct., 2016.

24.  D. Li, and S. K. Jayaweera, “Machine-learning aided optimal customer decision for an interactive smart grid,” IEEE Systems Journal, vol. 9, no. 4, pp.1529–1540, Dec., 2015.

25.  M. J. Ghorbani, M. A. Choudhry, and A. Feliachi, “A multiagent design for power distribution systems automation,” IEEE Transactions on Smart Grid, vol. 7, no. 1, pp. 329–339, Jan., 2016.

26.  Y. B. He, G. J. Mendis, and J. Wei, “Real-time detection of false data injection attacks in smart grid: a deep learning-based intelligent mechanism,” IEEE Transactions on Smart Grid, vol. 8, no. 5, pp. 2505–2516, Sep., 2017.

27.  G. K. Venayagamoorthy, R. K. Sharma, P. K. Gautam et al., “Dynamic energy management system for a smart microgrid,” IEEE Transactions on Neural Networks and Learning Systems, vol. 27, no.8, pp. 1643–1656, Aug., 2016.

28.  R. Thapa, L. Jiao, B. J. Oommen and A. Yazidi, "A Learning Automaton-Based Scheme for Scheduling Domestic Shiftable Loads in Smart Grids," IEEE Access, vol. 6, pp. 5348-5361, 2018.

29.  P. Palensky, D. Dietrich, "Demand side management: Demand response intelligent energy systems and smart loads", IEEE Trans. Ind. Informat., vol. 7, no. 3, pp. 381-388, Aug., 2011.

30.    S. H. Jung and J. C. Kim and C. B. Sim, “A Novel Data Prediction Model using Data Weights and Neural Network based on R for Meaning Analysis between Data”, Journal of the Korean multimedia society, vol. 18, no. 4, pp. 524-532, 2015.

31.    Martínez-Parrales R, Fuerte-Esquivel C.R, "A new unified approach for the state estimation and bad data analysis of electric power transmission systems with multi-terminal VSC-based HVDC networks," Journal of electric power systems research, vol. 160, pp .251-260, 2018.

32.    H. S. Cho, T. Yamazaki, M.S. Hahn, "AERO: extraction of user's activities from electric power consumption data," IEEE Transactions on Consumer Electronics, vol. 56, no. 3, pp. 2011-2018, 2010.

33.    S. H. Jung and C. S. Shin and Y. Y. Cho and J. W. Park and M. H. Park and Y. H. Kim and S. B. Lee and C.B. Sim, "Analysis Process based on Modify K-means for Efficiency Improvement of Electric power Data Pattern Detection," Journal of the Korean multimedia society, vol. 20, no. 12, pp. 1960-1969, 2017.

34.   Kroposki, B., Johnson, B., Zhang, Y., Gevorgian, V., Denholm, P., Hodge, B.M., Hannegan, B."Achieving a 100% Renewable Grid: Operating electric power Systems with Extremely High Levels of Variable Renewable Energy," IEEE Power Energy Magazine, vol. 15, no. 2, pp.61–73, 2017.

35.   Schwefel, H-P., et al. "Emerging Technologies Initiative ‘Smart Grid Communications’: Information Technology for Smart Utility Grids." 2018 IEEE International Conference on Communications, Control, and Computing Technologies for Smart Grids (SmartGridComm). IEEE, 2018.

36.   Wesoly, Malgorzata, Patrycja Ciosek, "Comparison of various data analysis techniques applied for the classification of pharmaceutical samples by electronic tongue." Sensors and Actuators B: Chemical 267 (2018): 570-580.

37.   Liébana-Cabanillas, Francisco, Francisco Muñoz-Leiva, and Juan Sánchez-Fernández. "A global approach to the analysis of user behavior in mobile payment systems in the new electronic environment." Service Business 12.1 (2018): 25-64.

38.    J. Mora-Florez, V. Barrera-Nuez, G. Carrillo-Caicedo, "Fault Location in Power Distribution Systems Using a Learning Algorithm for Multivariable Data Analysis", IEEE Transactions on Power Delivery, vol. 22, no. 3, pp. 1715-1721, 2007.

39.    S. H. Jung and C. S. Shin and Y. Y. Cho and J. W. Park and M. H. Park and Y. H. Kim and S. B. Lee and C.B. Sim, "A Novel of Data Clustering Architecture for Outlier Detection to Electric Power Data Analysis," KIPS Transactions on Software and Data Engineering, vol. 6, no. 10, pp. 465-472, Oct., 2017.

40.    M. H. Park and Y. H. Kim and S. B. Lee,, "A study on the Development of Energy IoT Platform", KIPS Tr. Comp. and Comm. Sys., vol. 5, no. 4, pp. 311-318, 2016.

41.    Seunghyeon Park, Sekyung Han, Yeongik Son, "Demand power forecasting with data mining method in smart grid," IEEE Innovative Smart Grid Technologies - Asia (ISGT-Asia), pp. 1-6, Dec. 4–7, 2017.

42.    Luo, Fengji, et al. "Cloud-based information infrastructure for next-generation power grid: Conception, architecture, and applications." IEEE Transactions on Smart Grid 7.4 (2016): 1896-1912.

43.    H., John A., and M. A. Wong, “Algorithm AS 136: A k-means clustering algorithm,” Journal of the Royal Statistical Society. Series C(Applied Statistics), vol. 28, no. 1, pp. 100-108, 1979.

44.    S. H. Jung, J. C. Kim, "CkLR Algorithm for Improvement of Data Prediction and Accuracy Based on Clustering Data," International Journal of Software Engineering and Knowledge Engineering, vol. 29, no. 05, pp. 631-652, 2019.

45.   Zhang, Geng, Chengchang Zhang, Huayu Zhang, "Improved K-means algorithm based on density Canopy." Knowledge-Based Systems 145 (2018): 289-297.

46.   Shrivastava, Puja, et al., "AKM—Augmentation of K-Means Clustering Algorithm for Big Data." Intelligent Engineering Informatics, Springer, Singapore, 2018. 103-109.

47.   Jia, Cangzhi, Yun Zuo,Quan Zou, "O-GlcNAcPRED-II: an integrated classification algorithm for identifying O-GlcNAcylation sites based on fuzzy undersampling and a K-means PCA oversampling technique." Bioinformatics 34.12 (2018): 2029-2036.

48.   Shakeel, P. Mohamed, et al., "Cloud based framework for diagnosis of diabetes mellitus using K-means clustering." Health information science and systems 6.1 (2018): 16.

49.   Ahlqvist, Emma, et al., "Novel subgroups of adult-onset diabetes and their association with outcomes: a data-driven cluster analysis of six variables." The lancet Diabetes & endocrinology 6.5 (2018): 361-369.

50.    J. Macqueen, “Some methods for classification and analysis of multivariate observations,” In Proceedings of Fifth Berkeley Symposium on Mathematical Statistics and Probability, vol. 1, University of California Press, pp. 281-297, 1967.

51.    K. Zhang, W. Bi, X. Zhang, X. Fu, K. Zhou, L. Zhu, “A New K-means Clustering Algorithm for Point Cloud,” J. of Hybrid Information Technology, vol.8, no.9, pp.157-170, 2015.

52.    F. Yuan, Z. H. Meng, H. X. Zhangz, C. R. Dong, “A New Algorithm to Get the Initial Centroids”, In Proceeding of the 3rd International Conference on Machine Learning and Cybernetics, pp. 26-29, 2004.

53.   S. H. Jung, J. C. Kim, "Efficiency Improvement of Classification Model Based on Altered K-Means Using PCA and Outlier," International Journal of Software Engineering and Knowledge Engineering, vol. 29, no. 5, pp.693–713, 2019.

54.    Christopher J. C. H. WatkinsPeter Dayan, "Q-Learning," Journal of Machine Learning, vol. 8, no. 3-4, pp. 279-292, 1992.

55.   Qingchen Zhang, Man Lin, Laurence T. Yang, Zhikui Chen, Samee U. Khan, Peng Li, "A double deep Q-learning model for energy-efficient edge scheduling." IEEE Transactions on Services Computing (2018).

56.   Van Hasselt, Hado, Arthur Guez, David Silver, "Deep reinforcement learning with double q-learning." Thirtieth AAAI Conference on Artificial Intelligence, 2016.

57.   Gu, S., Lillicrap, T., Sutskever, I., Levine, S., "Continuous deep q-learning with model-based acceleration," In International Conference on Machine Learning, 2016, June, pp.2829-2838.

58.   Mnih, V., Kavukcuoglu, K., Silver, D., Rusu, A. A., Veness, J., Bellemare, M. G., Petersen, S, "Human-level control through deep reinforcement learning," Nature, 518(7540), 529, 2015.

59.   Mnih, Volodymyr, et al., "Playing atari with deep reinforcement learning." arXiv preprint arXiv:1312.5602, (2013).

60.   Frank, Michael J., et al., "Genetic triple dissociation reveals multiple roles for dopamine in reinforcement learning." In Proceedings of the National Academy of Sciences, 104.41 (2007): 16311-16316.

61.    K. Arulkumaran, M. P. Deisenroth, M. Brundage, A. A. Bharath, "A Brief Survey of Deep Reinforcement Learning," IEEE Signal Processing Magazine, pp. 1-16, 2017.

62.    V. Mnih, A. P. Badia, M. Mirza, A. Graves, T. P. Lillicrap, T. Harley, D. Silver, K. Kavukcuoglu,“Asynchronous methods for deep reinforcement learning,” In International Conference on Machine Learning (ICML 2016), pp. 1928-1937, NY, USA, 2016.

63.    T. D. Kulkarni, A. Saeedi, S. Gautam, "Deep Successor Reinforcement Learning," arXiv preprint arXiv:1606.02396, pp. 1-10, 2016.

64.    Gregor, Karol, Danilo Jimenez Rezende, and Daan Wierstra. "Variational intrinsic control," arXiv preprint arXiv:1611.07507, 2016.

65.    G. Ostrovski, M. G. Bellemare, A. van de Oord, R. Munos, “Count-based exploration with neural density models,” In Proceedings of Machine Learning Research (PMLR 2017), arXiv preprint arXiv:1703.01310 2017.

66.    Jaderberg, M., Mnih, V., Czarnecki, W. M., Schaul, T., Leibo, J. Z., Silver, D., & Kavukcuoglu, K. "Reinforcement learning with unsupervised auxiliary tasks," arXiv preprint arXiv:1611.05397, 2016.

67.    K. Kansky, T. Silver, D. A. Mély, M. Eldawy, M. Lázaro-Gredilla, X. Lou, N. Dorfman, S. Sidor, S. Phoenix, D. George, "Schema Networks: Zero-shot Transfer with a Generative Causal Model of Intuitive Physics," arXiv:1706.04317, 2017.

68.    C. Fernando, D. Banarse, C. Blundell, Y. Zwols, D. Ha, A. A. Rusu, A. Pritzel, D. Wierstra "Pathnet: Evolution channels gradient descent in super neural networks," arXiv preprint arXiv:1701.08734, 2017.

69.  L. Yin, T. Yu, L. Zhou, L. Huang, X. Zhang, B. Zheng, "Artificial emotional reinforcement learning for automatic generation control of large-scale interconnected power grids," IET Gener. Transmiss. Distrib., vol. 11, no. 9, pp. 2305-2313, Jun., 2017.

70.  T. Yu, B. Zhou, K. W. Chan, L. Chen, B. Yang, "Stochastic optimal relaxed automatic generation control in non-markov environment based on multi-step Q(λ) learning", IEEE Trans. Power Syst., vol. 26, no. 3, pp. 1272-1282, Aug., 2011.

71.  Z. Yan and Y. Xu, "Data-Driven Load Frequency Control for Stochastic Power Systems: A Deep Reinforcement Learning Method With Continuous Action Search," IEEE Transactions on Power Systems, vol. 34, no. 2, pp. 1653-1656, Mar., 2019.

72.  D. Zhang X. Han and C. Deng, “Review on the Research and Practice of Deep Learning and Reinforcement Learning in Smart Grids,” Journal of Power And Energy Systems, Vol. 4, No. 3, pp.362-370 Sep., 2018.

73.  F. Ruelens, B. J. Claessens, S. Vandeal et al., “Residential demand response of thermostatically controlled loads using batch reinforcement learning,” IEEE Transactions on Smart Grid, vol. 8, no.5, pp. 2149–2159, Sep., 2017.

74.  Z. Wen, D. O’ Neill, and H. Maei, “Optimal demand response using device-based reinforcement learning,” IEEE Transactions on Smart Grid, vol. 6, no. 5, pp. 2312–2324, Sep. 2015.

75.    Leibo, D. Silver, and K. Kavukcuoglu. Reinforcement learning with unsupervised auxiliary tasks. In International Conference on Learning Representations, 2017.

76.    V. Mnih, K. Kavukcuoglu, D. Silver, A. A. Rusu, J. Veness, M. G. Bellemare, A. Graves, M. Riedmiller, A. K. Fidjeland, G. Ostrovski, S. Petersen, C. Beattie, A. Sadik, I. Antonoglou, H. King, D. Kumaran, D. Wierstra, S. Legg, and D. Hassabis,“Human-level control through deep reinforcement learning,” Nature, vol. 518, pp .529-533, Feb., 2015.

77.  Gang Ma, Linru Jiang, Guchao Xu, Jianyong Zheng, "A Model of Intelligent Fault Diagnosis of Power Equipment Based on CBR", Mathematical Problems in Engineering, vol. 2015, pp. 1, 2015.

78.  Connor Jennings, Dazhong Wu, Janis Terpenny, "Forecasting Obsolescence Risk and Product Life Cycle With Machine Learning," Components Packaging and Manufacturing Technology IEEE Transactions on, vol. 6, no. 9, pp. 1428-1439, 2016.

79.  P. Verma, P. Singh and R. D. S. Yadava, "Fuzzy c-means clustering based outlier detection for SAW electronic nose," 2017 2nd International Conference for Convergence in Technology (I2CT), Mumbai, 2017, pp. 513-519.

80.  M. Gupta, J. Gao, C.C. Aggarwal, J. Han, "Outlier detection for temporal data: A survey", IEEE Transactions on Knowledge and Data Engineering, vol. 26, pp. 2250-2267, 2014.

81.  W. Alves, D. Martins, U. Bezerra and A. Klautau, "A Hybrid Approach for Big Data Outlier Detection from Electric Power SCADA System," IEEE Latin America Transactions, vol. 15, no. 1, pp. 57-64, Jan., 2017.

82.  J. Xiong et al., "Enhancing Privacy and Availability for Data Clustering in Intelligent Electrical Service of IoT," IEEE Internet of Things Journal, vol. 6, no. 2, pp. 1530-1540, Apr., 2019.

83.  M. Salehi, C. Leckie, J. C. Bezdek, T. Vaithianathan and X. Zhang, "Fast Memory Efficient Local Outlier Detection in Data Streams," IEEE Transactions on Knowledge and Data Engineering, vol. 28, no. 12, pp. 3246-3260, 1 Dec., 2016.

84.  R. Menezes Salgado, T. Carvalho Machado and T. Ohishi, "Intelligent Models to Identification and Treatment of Outliers in Electrical Load Data," IEEE Latin America Transactions, vol. 14, no. 10, pp. 4279-4286, Oct., 2016.

85.    D .I. Park and S .H. Yoon, "Clustering and classification to characterize daily electricity demand," Journal of the Korean data & information science society, Vol. 28, No. 2, pp. 395-406, 2017.

86.    S. H. Ryu and H. S. Kim and D. E. Oh and J. K. No, "Customer Load Pattern Analysis using Clustering Techniques," KEPCO Journal on electric power and energy, Vol. 2, No. 1, pp. 61-69, 2016.

87.    Jin-Ho Shin, Bong-Jae Yi, Young-Il Kim, Heon-Gyu Lee, Keun Ho Ryu, "Spatiotemporal Load-Analysis Model for Electric Power Distribution Facilities Using Consumer Meter-Reading Data," IEEE Transactions on Power Delivery, Vol. 26, No. 2, pp. 736-743, 2011.
